# BAYESIAN WEAKS-TO-STRONG FROM TEXT CLASSIFICATION TO GENERATION

**Ziyun Cui**[1,2][*]**, Ziyang Zhang**[1,2][*]**, Guangzhi Sun**[3]**, Wen Wu**[2,3][†]**, Chao Zhang**[1,2][†]
[1]Department of Electronic Engineering, Tsinghua University, Beijing, China
[2]Shanghai Artificial Intelligence Laboratory, Shanghai, China
[3]Department of Engineering, University of Cambridge, Cambridge, UK
cui-zy24@mails.tsinghua.edu.cn, ziyang-z24@mails.tsinghua.edu.cn,
gs534@cam.ac.uk, wuwen@pjlab.org.cn, cz277@tsinghua.edu.cn

## ABSTRACT

Advances in large language models raise the question of how alignment techniques will adapt as models become increasingly complex and humans will only be able to supervise them weakly. Weak-to-Strong mimics such a scenario where weak model supervision attempts to harness the full capabilities of a much stronger model. This work extends Weak-to-Strong to WeakS-to-Strong by exploring an ensemble of weak models which simulate the variability in human opinions. Confidence scores are estimated using a Bayesian approach to guide the WeakS-to-Strong generalization. Furthermore, we extend the application of WeakS-to-Strong from text classification tasks to text generation tasks where more advanced strategies are investigated for supervision. Moreover, direct preference optimization is applied to advance the student model's preference learning, beyond the basic learning framework of teacher forcing. Results demonstrate the effectiveness of the proposed approach for the reliability of a strong student model, showing potential for superalignment. [1] [2]

## 1 INTRODUCTION

With the increase in computing power and the amount of training data available, the capabilities of large language models (LLMs) have been continuously brought closer to humans in many aspects. Despite their impressive performance, the preferences and values of pre-trained LLMs do not always align with humans, and dedicated approaches are needed to tackle the problem. Based on large-scale instruction datasets, supervised finetuning (SFT) encourages LLMs to follow human instructions more strictly and respond more safely (Wei et al., 2022). Reinforcement learning (RL) is commonly applied to such alignment. By collecting model output values and the corresponding human feedback, the model can be finetuned by RL to avoid generating undesirable outputs (Ziegler et al., 2019; Bai et al., 2022a; Ouyang et al., 2022; Nakano et al., 2021; Askell et al., 2021).

Since no current model has yet surpassed human intelligence, alignment methods, such as SFT and RL from human feedback (RLHF), remain effective. However, it is worthwhile considering future scenarios where artificial intelligence (AI) might surpass human intelligence in all aspects. Would the current alignment methods still be effective for such super AI models? How could humans supervise the super AI? To simulate this future scenario, an analogy situation is designed that downgrades both sides: using a weak model to simulate humans and a strong model to simulate future super AI (Burns et al., 2023), which is termed as *superalignment*. It has been demonstrated that adding a simple auxiliary loss can achieve effective Weak-to-Strong generalization, even if the weak model's supervision contains many errors, which offers hope of achieving superalignment. Nonetheless, this is just the beginning of exploring along the path of Weak-to-Strong.

---

[*]Equal contribution.
[†]Corresponding author.
[1]Supported by Shanghai Artificial Intelligence Laboratory.
[2]Code is available in https://github.com/cuiziyun/BayesianWS2S.

This paper extends the discussion on Weak-to-Strong in two directions. First, given the inherent capability gap between the weak model and the strong model, we propose using an ensemble of multiple weak models to improve the quality of weak supervision, which is called *WeakS-to-Strong*. This also accounts for the scenario where human opinions might diverge in tasks without a commonly accepted standard. Several approaches have been studied to effectively leverage the diversity of different weak models, and we adapt a Bayesian approach referred to as evidential deep learning (EDL) (Sensoy et al., 2018) to better estimate broader human preferences by learning a prior distribution over the weak labels produced by the weak models. Furthermore, the Weak-to-Strong task was primarily studied for text classification tasks (Burns et al., 2023). This paper extends the scope to text generation and compares three different approaches, Naive Multi-Weak, Joint Decoding and Bayesian Multi-Weak, as shown in Figure 1. The proposed Bayesian WeakS-to-Strong approach is demonstrated effective for both classification and generation. To better align with human preferences, a variant of direct preference optimization (DPO) (Rafailov et al., 2023) called conservative DPO (cDPO) (Eric, 2023) is used to finetune the strong model further on RL principles.

Our main contributions are summarized as follows.

- We proposed Bayesian WeakS-to-Strong which largely improves the quality of weak supervision and recovers the performance of the strong model.

- We propose to generalize both Weak-to-Strong and WeakS-to-Strong from text classification to generation tasks, extending their scope from content regulation to content generation.

- When applied to text generation, a token-level probability estimation is proposed to achieve soft labels for strong model training. We also propose the modified DPO algorithm under the Bayesian WeakS-to-Strong framework to further improve text generation performance.

## 2 RELATED WORK

**AI Alignment.** Aligning LLMs with human preferences has been a long-standing goal. Instruction tuning uses extensive datasets to improve LLMs' adherence to human instructions (Wei et al., 2022). RL allows LLMs to learn what types of responses humans prefer or dislike, with proximal policy optimization (PPO) being an effective RL method first applied to LLMs and becoming part of the standard RLHF process (Ziegler et al., 2019; Bai et al., 2022a; Ouyang et al., 2022; Nakano et al., 2021; Askell et al., 2021). However, PPO training can be unstable, leading to the development of DPO (Rafailov et al., 2023). Given the high cost of obtaining human preference data, researchers are now exploring the use of LLMs to simulate human preferences, provide feedback, and finetune models (Lee et al., 2023; Bai et al., 2022b; Gulcehre et al., 2023).

**Variability in Human Opinions.** In the process of aligning AI with human preferences, it is important to consider the inconsistency of human preferences (Liu et al., 2023), which often leads to multi-label problems. Previously, many approaches used simple methods like voting, aggregation, and averaging to handle multi-labels (Davani et al., 2022; Munos et al., 2023; Paun & Simpson, 2021; Prabhakaran et al., 2021). However, these methods do not effectively capture the preference differences of individual annotators included in the multiple labels. To better estimate the diversity of human preferences, Bayesian principles have been introduced. Deep learning models can be used to predict prior distributions, which are considered to produce the multiple available labels to estimate a broader range of human preferences (Sensoy et al., 2018; Wu et al., 2022; 2023).

**Weak-to-Strong.** The goal of Weak-to-Strong is to use a weak model to better supervise a strong model. OpenAI demonstrated that adding auxiliary confidence loss from the strong model itself can significantly improve the Weak-to-Strong performance (Burns et al., 2023). Following OpenAI's work, several studies emerged to introduce multiple weak models, used either in series or parallel, to improve the quality of supervision provided by the weak models (Liu & Alahi, 2024; Sang et al., 2024). Early model ensemble methods like Adaboost (Freund & Schapire, 1995) and Bootstrap aggregating (Leo, 1996) were explored in these works. Furthermore, confidence scores are incorporated to help the strong model assess the supervision quality provided by the weak models (Guo et al., 2024) and the weak model can be directly used to modify the output of the strong model (Ji et al., 2024).

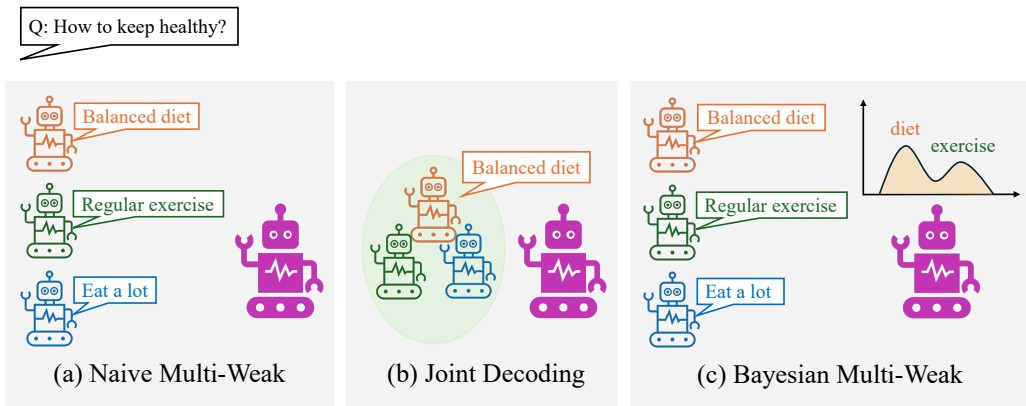

Figure 1: An overview diagram of the three ensemble approaches: (a) Naive Multi-Weak: directly learn all weak labels produced by weak models, (b) Joint Decoding: weak models collaboratively determine one single target, (c) Bayesian Multi-Weak: learn a prior distribution over weak labels.

## 3 WEAKS-TO-STRONG METHODOLOGY

### 3.1 PRELIMINARY: WEAK-TO-STRONG

The Weak-to-Strong pipeline (Burns et al., 2023) involves three steps: (i) create a weak supervisor by finetuning a small pre-trained model on ground-truth labels; (ii) train a strong student model $\boldsymbol{f_\Lambda}$ with weak supervision by finetuning a pre-trained LLM using "weak labels" generated by weak supervisors, where $\boldsymbol{\Lambda}$ is the parameters of the strong model; (iii) finetune the large pre-trained model directly using ground-truth labels which serve as the ceiling.

To leverage the superior generalization capabilities and prior knowledge of the strong model, a loss function with auxiliary confidence loss is proposed (Burns et al., 2023):

$$\mathcal{L} = (1-\gamma) \cdot \mathcal{L}_{\text{CE}}(\boldsymbol{f_\Lambda}(\boldsymbol{x}), \boldsymbol{y}_w) + \gamma \cdot \mathcal{L}_{\text{CE}}(\boldsymbol{f_\Lambda}(\boldsymbol{x}), \hat{\boldsymbol{f}}_\Lambda(\boldsymbol{x})) \tag{1}$$

where $\boldsymbol{y}_w$ represents the weak label from weak model, $\hat{\boldsymbol{f}}_\Lambda(\boldsymbol{x})$ refers to the predicted class of strong model given input $\boldsymbol{x}$, and $\mathcal{L}_{\text{CE}}(\cdot, \cdot)$ denotes the cross-entropy loss. The second term is an (optional) auxiliary self-training loss designed to increase the confidence of the strong model in itself. The weight of the second loss $\gamma$ linearly grows up from 0 to a pre-defined hyper-parameter $\gamma_{\max}$, which gradually reduces the weight on the weak labels and increases the weight on self-training when the number of training steps increases.

### 3.2 EXTENDING WEAK-TO-STRONG WITH MULTIPLE WEAK MODELS

Although it has been shown that the Weak-to-Strong approach can recover part of the strong model's performance (Burns et al., 2023), the errors in weak labels limit the performance of Weak-to-Strong generalization. In response to this problem, we propose to leverage the complementarity of the error patterns of multiple weak models using an ensemble strategy, which is referred to as WeakS-to-Strong.

A naive approach to implementing an ensemble of multiple weak models is to calculate the loss for each weak label respectively and then average these losses. An improvement of this approach is to take a weighted sum instead of a simple average:

$$\mathcal{L}_{\text{Naive}} = \sum_{i=1}^{N} \lambda_i \mathcal{L}_{\text{CE}}(\boldsymbol{f_\Lambda}(\boldsymbol{x}), \boldsymbol{y}_w^{(i)})), \tag{2}$$

where $N$ is the number of weak models, $\boldsymbol{y}_w^{(i)}$ is the $i$th weak label produced by the $i$th weak model, and $\lambda_i$ is a pre-defined weight of the loss regarding the the $i$th weak model. This approach is referred to as a Naive Multi-Weak system in the rest of the paper (as illustrated in Figure 1(a)), which is treated as one of the baselines.

### 3.3 BAYESIAN WEAKS-TO-STRONG

For superalignment, multiple weak models are used to mimic the subjective preferences of multiple humans, which can be considered as observations drawn from an underlying distribution of the opinions of all humans. The naive approach described in Section 3.2 solely relies on these observations. The number of observations (human annotations or weak labels) is often very limited due to the considerable cost of hiring a new human annotator or training a new weak model. Such a limited number of observations may not result in a good approximation of the true human opinion distribution. Having biased preferences or values is particularly unacceptable in the safety domain and can cause a failure of superalignment. Therefore, we propose a Bayesian WeakS-to-Strong approach based on EDL (Sensoy et al., 2018) to estimate the human opinion distribution based on the weak labels. Figure 1(c) illustrates the framework where three weak models are involved.

For a given input $x$, consider a weak label from the $i$th weak model $y_w^{(i)}$, which is a one-hot vector with $y_w^{(i,k)}$ being one if it belongs to class $k$ and zero otherwise. $y_w^{(i)}$ is sampled from a categorical distribution of weak labels $\mathrm{Cat}(\pi)$, where each component $\pi_k$ corresponds to the probability assignment over the possible classes $y_w^{(i)} \sim \mathrm{P}(y|\pi) = \mathrm{Cat}(\pi)$. EDL places a Dirichlet prior over the categorical distribution representing the probability of each categorical probability assignment, hence modelling second-order probability $\pi \sim \mathrm{p}(\pi|\alpha)$, where $\alpha$ is the hyperparameter of the Dirichlet prior. The strong model $f_\Lambda$ is trained to predict $\alpha$ for each input by minimizing the negative log-likelihood of sampling $y_w^{(i)}$ given the predicted Dirichlet prior:

$$\mathcal{L}_{\mathrm{NLL}}^{(i)} = -\log \int \mathrm{P}(y_w^{(i)}|\pi)\mathrm{p}(\pi|\alpha)\mathrm{d}\pi = \sum_{k=1}^{K} y_w^{(i,k)}(\log(\alpha_0) - \log(\alpha_k)), \tag{3}$$

where $K$ is the number of classes, $\alpha_0 = \sum_{k=1}^{K} \alpha_k$ is the Dirichlet strength, and $y_w^{(i,k)}$ is the $k^{th}$ value of label $y_w^{(i)}$. When a sample is not correctly classified, it is expected that the prior should approach non-informative prior for this sample. Following Sensoy et al. (2018), a regularization term $\mathcal{L}_{\mathrm{REG}}^{(i)}$ (KL-divergence between the misleading prediction and non-informative distribution, see Appendix H for details) is added to penalize incorrect predictions and calibrate uncertainty estimation, resulting in the final EDL loss $\mathcal{L}_{\mathrm{EDL}}^{(i)} = \mathcal{L}_{\mathrm{NLL}}^{(i)} + \lambda_{\mathrm{EDL}}\mathcal{L}_{\mathrm{REG}}^{(i)}$ where $\lambda_{\mathrm{EDL}}$ is the coefficient.

Apart from the class predicted by the weak models, the confidence of weak models is also incorporated for better distribution estimation. Let $(p_1^{(i)}, \ldots, p_K^{(i)})$ be the probability assignment predicted by the $i^{th}$ weak model, the EDL loss for each class is calculated based on the predicted probability assignment for each weak model and then combined in the same way as in Eqn. (2). That is,

$$\mathcal{L}_{\mathrm{EDL}}(f_\Lambda(x), \{y_w^{(i)}\}_{i=1}^{N}) = \sum_{i=1}^{N} \lambda_i \sum_{k=1}^{K} p_k^{(i)} \mathcal{L}_{\mathrm{EDL}}^{(i)}(f_\Lambda(x), \hat{y}_w^{(i,k)}) \tag{4}$$

where $\hat{y}_w^{(i,k)}$ is the predicted result, *i.e.*, one-hot vector for class $k$, and $\lambda_i$s are hyperparameters set to the same values as used in the Naive Multi-Weak approach. As a result, the auxiliary confidence loss described in Eqn. (1) is adapted for Bayesian WeakS-to-Strong as follows:

$$\mathcal{L} = (1 - \gamma) \cdot \mathcal{L}_{\mathrm{EDL}}(f_\Lambda(x), \{y_w^{(i)}\}_{i=1}^{N}) + \gamma \cdot \mathcal{L}_{\mathrm{EDL}}(f_\Lambda(x), \hat{f}_\Lambda(x)). \tag{5}$$

In the term of $\mathcal{L}_{\mathrm{EDL}}(f_\Lambda(x), \hat{f}_\Lambda(x))$, the class index predicted by the strong model $\hat{f}_\Lambda(x)$ is used as the target. That is to say, the predictions of the strong student model are applied as part of the distribution estimation along with the weak label.

## 4 WEAKS-TO-STRONG FOR SEQUENCE GENERATION

### 4.1 PROBABILITY ESTIMATION FOR WEAK SEQUENCE LABELS

To enable the strong model to directly generate trustworthy content rather than only being trained to understand whether the content is trustworthy or not, we propose to extend the scope of Weak-to-Strong from text classification to text generation. The key challenge of directly applying the

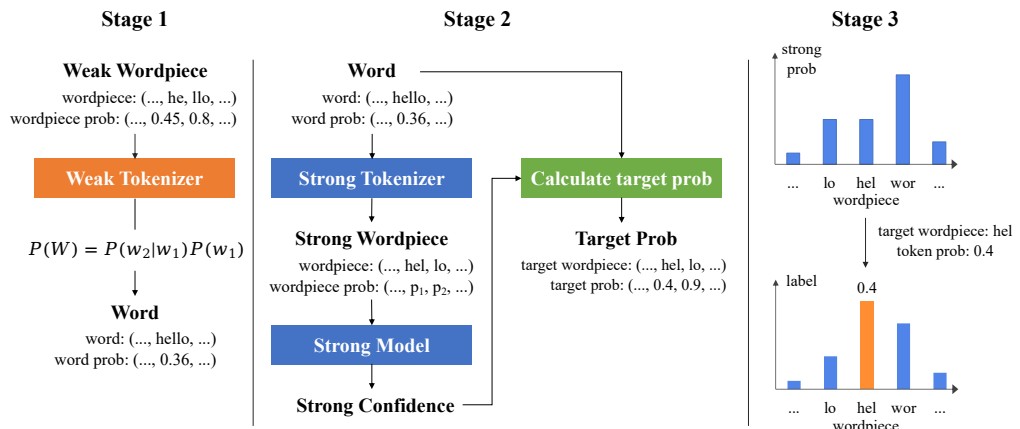

Figure 2: The process of transforming per-token confidence scores from the sequence tokenized by the weak model to the sequence tokenized by the strong. The word "hello" is used as an example. **Stage 1**: The words and word scores are obtained from the weak model wordpieces and their scores. **Stage 2**: The words are tokenized by the strong model tokenizer, and the tokenized sequences are fed into the strong model to obtain the strong model predicted probability (denoted as confidence) for each token $s_i$. This strong model confidence is then used to split word scores into target wordpiece probabilities $P(s_i)$ while keeping the probability of the word unchanged. **Stage 3**: The obtained target probability is transformed into the label. Probabilities of other categories are calculated by scaling the strong output distribution using $P(s_i)$.

Weak-to-Strong loss to the sequence generation task is the token-level soft labelling for the target sequence. As the tokenizers are different between weak and strong models, it is infeasible to obtain a one-to-one mapping from weak model output distributions to each token in the target sequence.

To obtain the soft label $\boldsymbol{y}_w$ for the strong model using weak model output probabilities and bridge the gap caused by different tokenizers, we use words as an intermediary, following the equation

$$\mathrm{P}(W) = \mathrm{P}(w_2|w_1)\mathrm{P}(w_1) = \mathrm{P}(s_2|s_1)\mathrm{P}(s_1), \tag{6}$$

where we use a word containing two wordpieces from both weak and strong tokenizers as an example, and $w_1, w_2$ and $s_1, s_2$ are both token strings that can form word $W$, which are generated by the tokenizers of the weak and strong models respectively. Figure 2 shows an example of the process in three stages. In stage 1, the per-token output probabilities of weak models are obtained when generating output sequences. The probabilities of wordpieces in a word are then multiplied together to obtain the score of word $W$, following Eqn. (6).

In stage 2, the word probability is used to assign probabilities to tokens from the strong model tokenizer. In the process of training the strong model via teacher-forcing, the model gives a probability to each token by applying softmax to the output logits. This probability can be seen as the model's confidence in predicting the target token, where we assume that the weak model and strong model have similar confidence for wordpiece tokens in similar positions. This allows us to to obtain the actual assignment of scores to each target token $s_i$, instead of assigning equal probabilities to all tokens involved. Taking an example of splitting a word $W$ into two target wordpieces $s_1$ and $s_2$, the decomposition can be approximated by

$$\log \mathrm{P}(s_1) = \frac{e^{-C_s(s_1)}}{e^{-C_s(s_1)} + e^{-C_s(s_2)}} \log \mathrm{P}(W), \quad \log \mathrm{P}(s_2) = \frac{e^{-C_s(s_2)}}{e^{-C_s(s_1)} + e^{-C_s(s_2)}} \log \mathrm{P}(W), \tag{7}$$

where $C_s(s_i)$ is strong model confidence at the step predicting wordpiece $s_i$ that is the maximum probability in the strong model output distribution at that step in practice. In this way, lower target probabilities are allocated to tokens with lower strong model confidence, while higher probabilities are allocated to tokens with higher strong model confidence.

After obtaining the probability of the target token of the strong model, the probabilities of other categories can be obtained by scaling strong prediction probabilities, which can be treated as the soft label, as shown in Figure 2. Then the obtained soft labels can be handled using methods similar to those used in classification (as described in Section 3). Notably, during the computation of EDL loss,

the sparsity caused by high dimensional spaces results in a large KL (Kullback–Leibler) penalty term. To solve this problem, a coefficient is added to the KL penalty to balance it with the magnitude of the negative log-likelihood term. Additionally, clamping is applied to restrict all values within an appropriate range, preventing potentially extremely large outliers on any particular token.

## 4.2 DPO for sequence generation optimization

Different from classification tasks, sequence generation tasks often benefit from sequence-level objectives that directly optimize the entire sequence jointly rather than the individual tokens separately. To further improve the strong model for sequence generation, direct preference optimization (DPO) (Rafailov et al., 2023) is investigated for WeakS-to-Strong after supervised finetuning, where we propose to use weak models to provide the preference for the strong model generation.

After the strong model is pretrained by supervised finetuning, it generates $M$ output sequences based on a given output. Then for each sequence, $N$ scores are computed by generating it using $N$ weak models separately (via teacher forcing) and aggregating the output (log-)probabilities. A weighted sum of the $N$ scores is performed as the final score assigned to each sequence. The sequence with the highest final score is viewed as the preferred sequence in DPO training, as shown below

$$\boldsymbol{y}_c = \underset{m,m=1,2,...,M}{\arg\max} \mathrm{P}(\boldsymbol{y}_s(m)) = \underset{m,m=1,2,...,M}{\arg\max} \sum_{i=1}^{N} \lambda_i \mathrm{P}(\boldsymbol{y}_s(m)|\boldsymbol{\theta}_i), \tag{8}$$

where $\boldsymbol{y}_s(m)$ is the $m$th output sequence generated by the strong model, $\mathrm{P}(\boldsymbol{y}_s(m)|\boldsymbol{\theta}_i)$ is computed using the $i$th weak model with model parameters $\boldsymbol{\theta}_i$, and $\lambda_i$ is the weight assigned to the $i$th weak model. The dispreferred sequence can be computed similarly by $\boldsymbol{y}_r = \arg\min \mathrm{P}(\boldsymbol{y}_s(m))$.

Considering potential errors by weak models, a variant of DPO, conservative DPO (cDPO) (Eric, 2023) with a more conservative target distribution is applied in our work. The loss of cDPO is

$$\mathcal{L}_{\mathrm{DPO}}^{\epsilon} = (1-\epsilon)\mathcal{L}_{\mathrm{DPO}}(\boldsymbol{\Lambda}, \boldsymbol{y}_c, \boldsymbol{y}_r) + \epsilon\mathcal{L}_{\mathrm{DPO}}(\boldsymbol{\Lambda}, \boldsymbol{y}_r, \boldsymbol{y}_c), \tag{9}$$

where $\epsilon$ is a small constant probability that labels are flipped to make DPO more conservative, and $\mathcal{L}_{\mathrm{DPO}}$ is the standard DPO loss in (Rafailov et al., 2023), which can be written as

$$\mathcal{L}_{\mathrm{DPO}}(\boldsymbol{\Lambda}, \boldsymbol{y}_c, \boldsymbol{y}_r) = -\log\sigma\Big(\beta\log\frac{\boldsymbol{f}_{\boldsymbol{\Lambda}}(\boldsymbol{y}_c)}{\boldsymbol{f}_{\mathrm{ref}}(\boldsymbol{y}_c)} - \beta\log\frac{\boldsymbol{f}_{\boldsymbol{\Lambda}}(\boldsymbol{y}_r)}{\boldsymbol{f}_{\mathrm{ref}}(\boldsymbol{y}_r)}\Big). \tag{10}$$

## 5 Experimental Setup

### 5.1 Datasets

**Classification Task.** The setup of the classification task follows Burns et al. (2023). The SciQ dataset (Welbl et al., 2017) is used, which contains 13,679 crowdsourced science exam questions about Physics, Chemistry and Biology, among others. The questions are in multiple-choice format with 4 answer options each. In our experiment, 5k data samples were extracted for training weak models and another 5k samples were reserved for generating weak labels to train the strong model. The standard test set which contains 1k data samples was used for the test. The data is restructured into a balanced binary classification task, *i.e.,* given a question and an answer, the model is required to determine whether the answer is correct.

**Slot filling.** The performance of WeakS-to-Strong on the reliability of generated content was evaluated on the slot-filling task, which is a crucial spoken language understanding task aiming at filling in the correct value for predefined slots (*e.g.* restaurant and hotel names). SLURP dataset (Bastianelli et al., 2020) was used which contains 16.5k sentences and 72k audio recordings of single-turn user interactions with a home assistant, annotated with scenarios, actions and entities. Only the reference transcriptions of the speech were used for training. Following Sun et al. (2023a;b), we designed the prompt with slot keys and descriptions in the same way. In our setup, 2k utterances from the train split were extracted for training the weak models, and another 2k utterances were reserved for generating weak labels and training the strong model. We report the performance of both weak and strong models on the standard SLURP test set.

## 5.2 MODELS AND BASELINES

**Models.** For classification, the Qwen-7B (Bai et al., 2023) model was applied as the strong student model. Five models were used as weak teachers: GPT2-Large (Radford et al., 2019), OPT-1.3B (Zhang et al., 2022), Pythia-1.4B (Biderman et al., 2023), BLOOM-1B1 (Le Scao et al., 2022), and TinyLlama_v1.1 (Zhang et al., 2024). The last linear layer which maps the embeddings to tokens is replaced with a linear classification head with two outputs to adapt language models to the classification setting.

For slot filling, Llama-2-7B (Touvron et al., 2023) was used as the strong model which yielded better performance than Qwen-7B in this task. The same set of weak models was used as the classification task. As before, both weak and strong models are finetuned with all model parameters.

Experiments with different numbers of weak models were conducted. One set involved three weak models (GPT2-Large, OPT-1.3B and Pythia-1.4B) in a WeakS-to-Strong experiment, which is referred to as *WeakS-to-Strong-3*. Another set included all five weak models, called *WeakS-to-Strong-5*.

**Baselines.** The proposed Bayesian WeakS-to-Strong method is compared to the following three baselines.

*Naive Multi-Weak.* The Naive Multi-Weak approach introduced in Section 3.2 servers as the baseline for both classification and generation tasks. For the classification task, the loss can be computed following Eqn. (2). For generation task, Eqn. (2) is modified as follows:

$$\mathcal{L}_{\text{Naive}}^{\text{Gen}} = \sum_{i=1}^{N} \lambda_i \frac{1}{T_i} \sum_{j=1}^{T_i} \mathcal{L}_{\text{CE}}(\boldsymbol{f}_{\boldsymbol{\Lambda}}(\boldsymbol{x}, \boldsymbol{y}_w^{(i,1)}, \ldots, \boldsymbol{y}_w^{(i,j-1)}), \boldsymbol{y}_w^{(i,j)})), \tag{11}$$

where $\{\boldsymbol{y}_w^{(i,1)}, \boldsymbol{y}_w^{(i,2)}, \ldots, \boldsymbol{y}_w^{(i,T)}\}$ is the sequence generated by $i$th weak model with length $T_i$. In contrast to the classification task, where the strong model $\boldsymbol{f}_{\boldsymbol{\Lambda}}$ takes only $\boldsymbol{x}$ as input, in the generation task, $\boldsymbol{x}$ is paired with the sequence generated by each weak model (which is the weak target) and then fed into the strong model to obtain predictions. Teacher-forcing is used during training.

*FlyingSquid.* FlyingSquid (Fu et al., 2020) is a method for weak supervision, estimating the accuracies and correlations among multiple noisy label functions (different weak labels in our case) without ground-truth data. Latent variable probabilistic graphical models are used to model these dependencies, with weak labels as observed variables and unobserved ground-truth labels as hidden variables. Since FlyingSquid is designed for binary classification, this baseline is only used for classification. Through this method, we get a label model with multiple weak labels and obtain the probability for the positive category, which is then used as a soft label in strong model training.

*Joint Decoding.* Joint Decoding is used as an additional baseline for text generation tasks, which is specifically designed for multiple weak model generations. In contrast to the Naive Multi-Weak scheme where each weak model provides a weak target sequence, Joint Decoding employs multiple weak models to collaboratively determine one single target, as illustrated in Figure 1(b). Specifically, we perform Joint Decoding in a re-ranking fashion. For each weak model, the top $M$ sequences are generated by beam search in decoding. The sequences from the $N$ weak models are gathered to form a list of $M \times N$ sequences. Then each sequence is scored in the same way in Section 4.2. The sequence with the highest final score is used as the target sequence for strong model training. Unless otherwise mentioned, $M = 5$ was used in the experiments.

Each weak-to-strong experiment was run three times with different random seeds. The average and standard deviation were reported. More implementation details can be found in Appendix C.

## 5.3 EVALUATION METRICS

The classification task is evaluated by accuracy, and the SLU-F1 (Bastianelli et al., 2020) is used for slot filling, which combines both word-level and character-level F1 scores to give partial credit to non-exact match predictions. Performance gap recovered (PGR) (Burns et al., 2023) is used to measure the performance gap recovered with weak supervision, which is defined as $(\mathcal{P} - \mathcal{P}_w)/(\mathcal{P}_s - \mathcal{P}_w)$ where $\mathcal{P}$ is Weak-to-Strong performance, $\mathcal{P}_s$ strong performance and $\mathcal{P}_w$ weak performance. For multiple weak model cases, the average of PGRs for each weak model is treated as the final result.

Table 1: Performance of single model on text classification task. Trained by ground-truth labels.

|  | Pre-trained model | # Param | Accuracy |
|---|---|---|---|
| Strong Model (ceiling) | Qwen-7B | 7.7B | 0.898 |
| Weak Model | GPT2-Large | 0.8B | 0.717 |
|  | OPT-1.3B | 1.3B | 0.699 |
|  | Pythia-1.4B | 1.4B | 0.685 |
|  | BLOOM-1B1 | 1.1B | 0.729 |
|  | TinyLlama_v1.1 | 1.1B | 0.731 |

Table 2: Weak(S)-to-Strong performance (train the strong model on the weak label) on text classification task for with (w/) and without (w/o) auxiliary loss. $\gamma$ in Eqn. (1) and Eqn. (5) is set to 0 if auxiliary loss is not used. Experiments with 3 weak models and 5 weak models were conducted. Each experiment was run with three different random seeds. The best results are shown in bold.

|  |  | w/o aux loss | | w/ aux loss | |
|---|---|---|---|---|---|
|  |  | Accuracy | PGR | Accuracy | PGR |
| Weak-to-Strong | GPT2-Large | $0.808 \pm 0.007$ | $0.503 \pm 0.034$ | $0.828 \pm 0.006$ | $0.614 \pm 0.029$ |
|  | OPT-1.3B | $0.807 \pm 0.005$ | $0.541 \pm 0.026$ | $0.841 \pm 0.012$ | $0.714 \pm 0.058$ |
|  | Pythia-1.4B | $0.775 \pm 0.009$ | $0.421 \pm 0.042$ | $0.793 \pm 0.009$ | $0.507 \pm 0.042$ |
|  | BLOOM-1B1 | $0.823 \pm 0.015$ | $0.556 \pm 0.087$ | $0.843 \pm 0.008$ | $0.677 \pm 0.049$ |
|  | TinyLlama_v1.1 | $0.832 \pm 0.004$ | $0.603 \pm 0.024$ | $0.838 \pm 0.005$ | $0.643 \pm 0.030$ |
| WeakS-to-Strong-3 | Naive Multi-Weak | $0.816 \pm 0.002$ | $0.586 \pm 0.008$ | $0.831 \pm 0.013$ | $0.661 \pm 0.064$ |
|  | FlyingSquid | $0.809 \pm 0.005$ | $0.549 \pm 0.026$ | $0.825 \pm 0.003$ | $0.631 \pm 0.013$ |
|  | Bayesian Mutli-Weak | $0.819 \pm 0.006$ | $0.600 \pm 0.033$ | $\mathbf{0.850} \pm 0.006$ | $\mathbf{0.756} \pm 0.028$ |
| WeakS-to-Strong-5 | Naive Multi-Weak | $0.832 \pm 0.005$ | $0.641 \pm 0.025$ | $0.853 \pm 0.006$ | $0.754 \pm 0.032$ |
|  | FlyingSquid | $0.832 \pm 0.004$ | $0.643 \pm 0.023$ | $0.855 \pm 0.007$ | $0.768 \pm 0.035$ |
|  | Bayesian Mutli-Weak | $0.831 \pm 0.008$ | $0.627 \pm 0.027$ | $\mathbf{0.866} \pm 0.006$ | $\mathbf{0.828} \pm 0.038$ |

# 6 RESULTS

## 6.1 TEXT CLASSIFICATION

The proposed Bayesian WeakS-to-Strong approach was first evaluated on a classification task. Table 1 shows the respective performance of the strong model and the weak models trained using ground-truth labels, with the former being the ceiling of the Weak(S)-to-Strong approaches. The strong model has about 7 times the number of parameters as the weak models, which also leads to about 28% relative improvement in the classification accuracy. Results of Weak(S)-to-Strong approaches are shown in Table 2. $\gamma$ in Eqn. (1) and Eqn. (5) were set to 0 if auxiliary loss was not used. It can be seen that auxiliary loss was effective for both. Comparing single Weak-to-Strong results, different weak models show significant differences in performance. For example, Pythia-1.4B recovered only 50% of the strong performance, while OPT-1.3B recovered around 70%.

The Naive Multi-Weak baseline, FlyingSquid approach and Bayesian Multi-Weak using EDL were applied to the classification task for WeakS-to-Strong. Experiments with three weak models (GPT2-Large, OPT-1.3B and Pythia-1.4B) and all five weak models were conducted. With three weak models, the Naive Multi-Weak method got an average PGR of 0.661, slightly outperforming FlyingSquid method but still lower than the single OPT-1.3B model, and Bayesian Multi-Weak boosted PGR to 0.756. This indicates that a naive ensemble approach doesn't necessarily outperform the best single model, especially when there is a certain weak model which does not perform well. The Bayesian approach can increase the fault tolerance with the usage of prior in this case, as it learns patterns from the entire dataset. With five weak models compared to three, the Bayesian Multi-Weak approach further increased average PGR to 0.828, which is 16% relatively higher than the best single model and 8% relatively higher than baselines, and consistently better with all seeds. The results show the effectiveness of the Bayesian approach for distribution estimation.

Table 3: Performance of single model on text generation task. Trained by ground-truth labels.

| | Pre-trained model | # Param | SLU-F1 |
|---|---|---|---|
| Strong Model (ceiling) | Llama-2-7B | 6.7B | 0.748 |
| Weak Model | GPT2-Large | 0.8B | 0.660 |
| | OPT-1.3B | 1.3B | 0.665 |
| | Pythia-1.4B | 1.4B | 0.680 |
| | BLOOM-1B1 | 1.1B | 0.651 |
| | TinyLlama_v1.1 | 1.1B | 0.676 |

Table 4: Weak(S)-To-Strong performance on text generation task, training strong model on weak labels, for with (w/) and without (w/o) auxiliary loss. In cases without auxiliary loss, $\gamma$ in Eqn. (1) and Eqn. (5) is set to 0. Experiments were conducted using 3 and 5 weak labels, each run with three different random seeds. The best results are shown in bold.

| | | w/o aux loss | | w/ aux loss | |
|---|---|---|---|---|---|
| | | SLU-F1 | PGR | SLU-F1 | PGR |
| Weak-to-Strong | GPT2-Large | $0.687 \pm 0.011$ | $0.303 \pm 0.125$ | $0.673 \pm 0.012$ | $0.150 \pm 0.139$ |
| | OPT-1.3B | $0.660 \pm 0.059$ | $-0.066 \pm 0.715$ | $0.696 \pm 0.009$ | $0.367 \pm 0.103$ |
| | Pythia-1.4B | $0.702 \pm 0.007$ | $0.320 \pm 0.095$ | $0.691 \pm 0.022$ | $0.173 \pm 0.322$ |
| | BLOOM-1B1 | $0.690 \pm 0.021$ | $0.399 \pm 0.220$ | $0.684 \pm 0.021$ | $0.337 \pm 0.220$ |
| | TinyLlama_v1.1 | $0.667 \pm 0.006$ | $0.000 \pm 0.084$ | $0.658 \pm 0.016$ | $-0.250 \pm 0.224$ |
| WeakS-to-Strong-3 | Naive Multi-Weak | $0.711 \pm 0.008$ | $0.531 \pm 0.101$ | $0.694 \pm 0.013$ | $0.318 \pm 0.169$ |
| | Joint Decoding | $0.703 \pm 0.003$ | $0.434 \pm 0.032$ | $0.704 \pm 0.015$ | $0.442 \pm 0.191$ |
| | Bayesian Mutli-Weak | $0.712 \pm 0.014$ | $0.549 \pm 0.180$ | $\mathbf{0.714} \pm 0.014$ | $\mathbf{0.574} \pm 0.176$ |
| WeakS-to-Strong-5 | Naive Multi-Weak | $0.716 \pm 0.010$ | $0.606 \pm 0.123$ | $0.694 \pm 0.012$ | $0.328 \pm 0.144$ |
| | Joint Decoding | $0.671 \pm 0.010$ | $0.037 \pm 0.120$ | $0.675 \pm 0.005$ | $0.091 \pm 0.066$ |
| | Bayesian Mutli-Weak | $0.718 \pm 0.014$ | $0.627 \pm 0.173$ | $\mathbf{0.721} \pm 0.013$ | $\mathbf{0.668} \pm 0.166$ |

## 6.2 TEXT GENERATION

For the text generation task on slot filling, the performance of the student strong model and teacher weak model finetuned on ground-truth labels is presented in Table 3. The strong ceiling performance is 0.748, and the highest weak performance is 0.680.

The Weak(S)-to-Strong performances are reported in Table 4. For a single weak model, the Weak-to-Strong model performance didn't necessarily surpass the original weak performance (*e.g.* TinyL-lama_v1.1), and the highest PGR is 0.399. With five weak models, our proposed Bayesian Multi-Weak approach achieved an average PGR of 0.668, which is 26% better than a single weak model, 6% better than the naive baseline, and consistently better across three seeds. Comparing WeakS-to-Strong performance with and without auxiliary loss, adding auxiliary loss is not effective for naive approaches, but improves the performance of Bayesian approaches. It indicates that the proposed Bayesian method, where the predictions of a strong model are applied as part of the distribution estimation as in Eqn. (5), more effectively integrates strong model predictions into the training process, showing the effectiveness of our Bayesian approach for the model ensemble. The ablation study can be found in Appendix E. A noticeable decline has been observed for Joint Decoding when the number of weak models increases from three to five. This may result from the poor performance of the TinyLlama weak model which affects the quality of the generated weak target. Recall that in Joint Decoding, multiple weak models collaboratively determine one single target. In contrast, Naive Multi-Weak and Bayesian Multi-Weak methods still benefit from the increased number of weak models, which indicates that they are more robust against the quality of a single weak model. Additional experiments on Joint Decoding can be found in Appendix G for more analysis and insights.

Based on the strong model supervised by five weak models on the Bayesian Multi-Weak approach with auxiliary loss, a cDPO training is conducted, which further train the model in student-forcing form. Three separate cDPO experiments were conducted using different initial models obtained in SFT stage with three different seeds. The results are shown in Table 5. After cDPO, three models

Table 5: Results before and after DPO. Based on the Bayesian Multi-Weak model using five weak models (the WeakS-to-Strong-5 setting) with auxiliary loss. Average results on three seeds are reported. Note that for different seeds, the initial SFT models are different.

| | seed-0 | | seed-1 | | seed-2 | | Average | |
|---|---|---|---|---|---|---|---|---|
| | SLU-F1 | PGR | SLU-F1 | PGR | SLU-F1 | PGR | SLU-F1 | PGR |
| Before cDPO | 0.706 | 0.477 | 0.730 | 0.776 | 0.728 | 0.751 | $0.721 \pm 0.013$ | $0.668 \pm 0.166$ |
| After cDPO | 0.707 | 0.490 | 0.733 | 0.813 | 0.733 | 0.813 | $0.724 \pm 0.015$ | $0.705 \pm 0.187$ |

showed consistent performance improvements. The average PGR reached **0.705**, 6% relatively better than that before DPO, with a maximum PGR of **0.813**.

### 6.3 COMPLEMENTARITY OF WEAK MODELS

An experiment about the complementarity of different weak models was conducted. For classification models, the agreement is assessed by calculating the accuracy of each model's predictions on test set, treating outputs from other models as references. For generation models, the Levenshtein distance is calculated between different outputs from two models for a same input, which is obtained using the minimum edit number required to change one sequence to another. The average Levenshtein distance across all samples in the test set is used to measure the agreement between two models.

The results are shown in Figure 3. The agreement among different weak models is among 0.75 for classification models, while for generation task the maximum agreement between different models is 0.56. This suggests that the consistency among different weak models is low, thus they can complement each other well as the faults made by different weak models are not the same. Moreover, comparing with Naive Multi-Weak approach, our proposed Bayesian method estimates a distribution based on weak labels, learning patterns from the entire dataset, thereby increasing the tolerance for fault in weak models, as shown in results in Section 6.1 and 6.2.

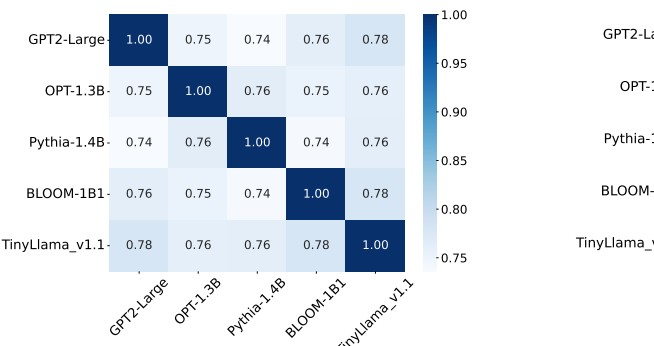
(a) Agreement of 5 models on classification.

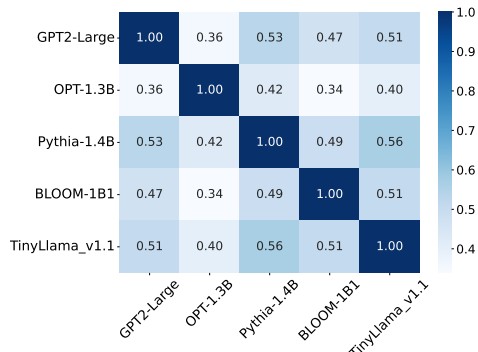
(b) Agreement of 5 models on generation.

Figure 3: Agreement of weak models. The similarity between classification models was assessed by calculating the accuracy of each model's predictions against the others on the test set. For generation models, the agreement is obtained through the Levenshtein distance.

## 7 CONCLUSION

This paper extends the Weak-to-Strong framework to WeakS-to-Strong by leveraging an ensemble of weak models to capture the variability in human opinions. We propose a Bayesian inference method, Bayesian WeakS-to-Strong, to more accurately estimate the weak label distribution based on the outputs of multiple weak models. Additionally, while the original Weak-to-Strong method was limited to text classification tasks, this paper expands its applicability to text generation, enabling both the assessment of content trustworthiness and the generation of trustworthy content. Finally, DPO is utilized to enhance the student model's preference for learning, going beyond the traditional teacher-forcing approach. Our results demonstrate the effectiveness of Bayesian WeakS-to-Strong for both classification and generation tasks, highlighting its potential for superalignment.

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

# A    LIMITATIONS

The proposed Bayesian WeakS-to-Strong method was tested on two different types of tasks: text classification and generative slot filling. We believe the proposed method is general, and further experiments on other applications are reserved for future work. Due to computational resource limitations, experiments on only three and five weak models were conducted in this paper, mimicking the situation where human annotations are costly and time-consuming to obtain. We believe that the capabilities of the strong model can be recovered to a greater extent when more weak models are involved.

# B    BROADER IMPACT

As an approach that enhances the original Weak-To-Strong method, our paper will have the following positive broader impact:

- By ensuring strong LLMs behave in ways that are predictable and consistent with societal values, the proposed WeakS-To-Strong further increase public trust in AI technologies.
- The use of weak model ensemble for strong model training helps to reduce risks of ethical violations such as gender or racial biases.
- Multiple weak models can be more easily updated or replaced to adapt to changing social norms and values. This flexibility allows LLMs to remain relevant and responsive to societal changes, ensuring that they continue to serve the public good over time.

This paper does not give rise to any additional potential biases beyond the ones directly inherited from the pre-trained LLM checkpoints. We encourage the practitioner to carefully select weak models such that the biases present in individual weak models do not accumulate or amplify when combined.

# C    IMPLEMENTATION DETAILS

All models were trained on NVIDIA A800 GPUs using the bfloat16 data type. For the classification tasks, the Adam optimizer was used with a cosine learning rate scheduler and no warm-up period. The batch size was set to 32, with a mini-batch size of 1. The weak models were finetuned on the ground-truth labels with an initial learning rate of $5 \times 10^{-5}$, while the strong models were trained with a starting learning rate of $1 \times 10^{-5}$ (both on the weak labels and ground-truth labels). The Weak(S)-to-Strong training was run for two epochs. The weights for different weak models ($\lambda$) was set to average values to simplify the classification experiments (refer to Appendix F for a comparison between average and fixed weights).

For generation tasks, the AdamW optimizer was used with a linear learning rate scheduler, also with no warm-up. The initial learning rates were set at $4 \times 10^{-5}$ for GPT2-Large and Pythia-1.4B, and $8 \times 10^{-5}$ for OPT-1.3B, with a batch size of 8 (mini-batch size of 4). These models were trained for 15 epochs. The checkpoints with the lowest validation loss were selected to ensure the quality of weak labels produced by the weak models. The strong model was trained with a batch size of 2 (mini-batch size of 1) and an initial learning rate of $1 \times 10^{-5}$, evaluated at the end of two epochs. In the training for strong ceiling performance, the hyperparameters were adjusted based on the validation set. For Weak-to-Strong training, in which case the ground truth is not accessible, we aligned the hyperparameter settings with those used in strong ceiling training. The weights $\lambda$ in Eqn. 4 were set to (0.1, 0.3, 0.2, 0.3, 0.1) for GPT2-Large, OPT-1.3B, Pythia-1.4B, BLOOM-1B1 and TinyLlama_v1.1 respectively.

For DPO, the initial learning rate was set to $5 \times 10^{-7}$ for two epochs, with the cDPO's hyperparameter $\beta$ set to 2.0 and label smoothing $\epsilon$ as 0.1. Other settings remain the same as the generation tasks.

# D    EXPERIMENTS ON AN ADDITIONAL DATASET

To verify the generalization of our method, additional experiments were conducted on another dataset, CosmosQA (Huang et al., 2019), for the classification task. CosmosQA is a large-scale dataset of

problems that require commonsense-based reading comprehension. Like the preprocess on the SciQ dataset, 5k data samples were extracted for training weak models, another 5k samples for strong models, and 1k samples for testing. This dataset was also reformatted into a binary classification format (*i.e.*, determining correctness). Experiments with three weak models were conducted. The weak model performance and strong performance are listed in Table 6 and Weak(S)-to-Strong performance in Table 7.

Table 6: Performance of single model on text classification task on CosmosQA dataset. Trained by ground-truth labels.

|  | Pre-trained model | # Param | Accuracy |
|---|---|---|---|
| Strong Model (ceiling) | Qwen-7B | 7.7B | 0.847 |
| Weak Model | GPT2-Large | 0.8B | 0.642 |
|  | OPT-1.3B | 1.3B | 0.642 |
|  | Pythia-1.4B | 1.4B | 0.654 |

Table 7: Weak(S)-to-Strong performance on text classification task for with and without auxiliary loss on CosmosQA dataset. Experiments with 3 weak models were conducted. Each experiment was run with three different random seeds. The best results are shown in bold.

|  |  | w/o aux loss | | w/ aux loss | |
|---|---|---|---|---|---|
|  |  | Accuracy | PGR | Accuracy | PGR |
| Weak-to-Strong | GPT2-Large | $0.652 \pm 0.007$ | $0.077 \pm 0.032$ | $0.651 \pm 0.007$ | $0.073 \pm 0.032$ |
|  | OPT-1.3B | $0.687 \pm 0.005$ | $0.218 \pm 0.022$ | $0.704 \pm 0.009$ | $0.304 \pm 0.044$ |
|  | Pythia-1.4B | $0.685 \pm 0.010$ | $0.162 \pm 0.052$ | $0.731 \pm 0.002$ | $0.399 \pm 0.009$ |
| WeakS-to-Strong-3 | Naive Multi-Weak | $0.691 \pm 0.009$ | $0.232 \pm 0.043$ | $0.698 \pm 0.007$ | $0.267 \pm 0.034$ |
|  | FlyingSquid | $0.699 \pm 0.004$ | $0.272 \pm 0.020$ | $0.706 \pm 0.004$ | $0.303 \pm 0.030$ |
|  | Bayesian Mutli-Weak | $0.694 \pm 0.007$ | $0.247 \pm 0.033$ | $\mathbf{0.760} \pm 0.005$ | $\mathbf{0.571} \pm 0.025$ |

Table 6 shows that, on the CosmosQA dataset, both the weak and strong models perform worse compared to that on the SciQ dataset, with the weak model's performance dropping more. In the Weak(S)-to-Strong experiments, neither multi-weak baseline outperformed the best single model (Pythia-1.4B with auxiliary loss), while our Bayesian approach significantly exceeded it, showing the effectiveness of our method, especially when weak model performance varies.

## E   ABLATION STUDY ON PROBABILITY ESTIMATION FOR WEAK SEQUENCES

The importance of probability estimation for weak sequences was explored in this section. Two key steps were calculating target wordpiece probability using the word as a gap bridge, and estimating probabilities of other categories by scaling the strong output (see Section 4.1 for details). The results are shown in Table 8. The average PGR of using a one-hot label (Line 1) is negative, showing that WeakS-to-Strong performance doesn't surpass the weak model without probability estimation. Estimating probability for the target wordpiece significantly improved performance compared to using one-hot labels. This indicates the necessity of introducing target probabilities during training, which allows the strong student model to learn the weak model's confidence for its generated labels, considering it may be incorrect. Based on this, adding probabilities of other categories, the overall precision of target label estimation improves, resulting in about 20% PGR gain. This experiment highlights the importance of precise probability estimation for weak sequences.

## F   WEIGHTS OF WEAK MODELS

The selection for weights of weak models was explored in this section, where experiments on average, dynamic and fixed weights were conducted. For average, the weight $\lambda_i$ in Eqn. (2) and Eqn. (4) is set to the same value for each weak model. That is to say, the losses are calculated for each weak model respectively and then averaged as the final loss. For fixed weight, the weights are set to a fixed value based on the performance of different weak models (refer to C for specific value). For

Table 8: Ablation study on probability estimation for weak sequence. Tested whether to calculate the probability for target wordpiece, and the estimation for probabilities for other categories.

| Target wordpiece | Other category | SLU-F1 | PGR |
|:---:|:---:|:---:|:---:|
| | | $0.656 \pm 0.016$ | $-0.145 \pm 0.204$ |
| ✓ | | $0.704 \pm 0.025$ | $0.453 \pm 0.305$ |
| ✓ | ✓ | $0.721 \pm 0.013$ | $0.668 \pm 0.166$ |

dynamic weight, for a certain sample, the confidence in the utterance of a weak model is treated as a weight for that weak model, considering the importance of different weak models could vary on different samples. The results are shown in Table 9. Compared to the average, fixed weight improves the average PGR by about 8% because it assigns weights based on the performance of different weak models, rather than just averaging. Dynamic weight shows a noticeable decline in performance compared to fixed weight. It may be because our proposed per-token target probability already provided the student model with fine-grained information about the reliability of the label, making dynamic weight entirely redundant.

Table 9: Different weighting strategies for weak models: experiments using average, dynamic and fixed weights.

| | SLU-F1 | PGR |
|:---|:---:|:---:|
| Average | $0.715 \pm 0.019$ | $0.589 \pm 0.237$ |
| Dynamic weight | $0.687 \pm 0.013$ | $0.241 \pm 0.163$ |
| Fixed weight | $0.721 \pm 0.013$ | $0.668 \pm 0.166$ |

# G   PERFORMANCE OF JOINT DECODING

## G.1   THE IMPACT OF THE QUALITY OF WEAK MODELS

This section provides additional experiments where different weak model combinations are used: (i) three weak models (GPT2-Large, OPT-1.3B, Pythia-1.4B), (ii) four models (adding BLOOM-1B1); and (iii) all five models (further adding TinyLlama_v1.1). For TinyLlama_v1.1, two setups were investigated: (i) only using it to generate weak target sequence but not for scoring (scoring done by other four models); (ii) using it for both generation and scoring. The results are listed in Table 10. Compared to the three-weak-model system, introducing the 4th model shows a 10% increase in PGR. However, incorporating TinyLlama as the 5th weak model undermines the performance (last row in Table 10), even nearly fails to surpass weak model performance. A possible reason is that TinyLlama performs poorly on the text generation task. The impact of the TinyLlama can be reduced by excluding it from scoring. As shown in the second to the last row of Table 10, without TinyLlama scoring, the results with five weak models are close to those with three. This indicates that TinyLlama's poor scoring ability prevents it from selecting a good target sequence among the candidates and the poor quality from a certain weak model can largely impact the overall results for Joint Decoding methods. In contrast, as discussed in Section 6.2, the proposed Bayesian WeakS-to-Strong method is more robust against the quality of a single weak model.

## G.2   THE IMPACT OF BEAM SIZE

As introduced in Section 5.2, in Joint Decoding, each weak model generates $M$ output sequences by beam search in decoding. This section investigates Joint Decoding with different beam sizes $M$, as shown in Table 11. For WeakS-to-Strong-3, results with different $M$ yields similar results. However, for WeakS-to-Strong-5, with TinyLlama included in the weak model set, the result with $M = 10$ is significantly worse than that with $M = 3$ and $M = 5$. The experiments on beam size further demonstrate that the poor results of the WeakS-to-Strong-5 are due to its weak ability to select the target sequence, as $M = 10$ introduces more distractions than $M = 5$.

Table 10: Joint Decoding performance with different weak models. Experiments with three weak models (GPT2-Large, OPT-1.3B, Pythia-1.4B), four weak models (adding BLOOM-1B1), and all five weak models are conducted.

| 3 Weak models | BLOOM-1B1 | TinyLlama_v1.1 Generating | Scoring | SLU-F1 | PGR |
|:---:|:---:|:---:|:---:|:---:|:---:|
| ✓ | | | | $0.703 \pm 0.003$ | $0.434 \pm 0.032$ |
| ✓ | ✓ | | | $0.711 \pm 0.008$ | $0.549 \pm 0.100$ |
| ✓ | ✓ | ✓ | | $0.706 \pm 0.009$ | $0.452 \pm 0.086$ |
| ✓ | ✓ | ✓ | ✓ | $0.671 \pm 0.010$ | $0.037 \pm 0.120$ |

Table 11: Joint Decoding performance with different beam sizes in beam search when weak models generate labels. Both WeakS-to-Strong-3 and WeakS-to-Strong-5 are explored.

| | WeakS-to-Strong-3 | | WeakS-to-Strong-5 | |
| | SLU-F1 | PGR | SLU-F1 | PGR |
|:---:|:---:|:---:|:---:|:---:|
| M=3 | $0.706 \pm 0.012$ | $0.471 \pm 0.155$ | $0.669 \pm 0.023$ | $0.022 \pm 0.281$ |
| M=5 | $0.703 \pm 0.003$ | $0.434 \pm 0.032$ | $0.671 \pm 0.010$ | $0.037 \pm 0.120$ |
| M=10 | $0.703 \pm 0.004$ | $0.431 \pm 0.045$ | $0.650 \pm 0.004$ | $-0.224 \pm 0.050$ |

## H    REGULARISING TERM OF EDL

As introduced in Section 3.3, the negative log-likelihood of a sample $y$ with a predicted Dirichlet prior with hyperparameter $\alpha$ is:

$$\mathcal{L}_{\text{NLL}} = -\log \int \text{P}(\boldsymbol{y}_w|\boldsymbol{\pi})\text{p}(\boldsymbol{\pi}|\boldsymbol{\alpha})\text{d}\boldsymbol{\pi} = \sum_{k=1}^{K} y_w^{(k)}(\log(\alpha_0) - \log(\alpha_k))$$

When a sample is not correctly classified, it is expected the total evidence shrinks to zero for the sample. Taking this into consideration, Sensoy et al. (2018) added a regularization term to penalise the misleading evidence. The loss with this regularising term reads

$$\mathcal{L}_{\text{EDL}} = \mathcal{L}_{\text{NLL}} + \lambda_t \mathcal{L}_{\text{KL}}(\text{Dir}(\boldsymbol{\pi}|\tilde{\boldsymbol{\alpha}})||\text{Dir}(\boldsymbol{\pi}|\boldsymbol{1}))$$

where the KL term refers to the $\mathcal{L}_{\text{REG}}$ in Section 3.3, $\text{Dir}(\boldsymbol{\pi}|\boldsymbol{1})$ denotes a Dirichlet distribution with zero total evidence, $\tilde{\boldsymbol{\alpha}} = \boldsymbol{y} + (1 - \boldsymbol{y}) \odot \boldsymbol{\alpha}$ is the Dirichlet parameter after removal of the non-misleading evidence from predicted $\boldsymbol{\alpha}$, and $\lambda_t$ is the annealing coefficient. By adding a KL-divergence between the Dirichlet distribution with misleading evidence and zero total evidence, the total evidence is enforced to shrink to zero for the simple which is not correctly classified. The annealing coefficient increases by training step, enabling the model to explore the parameter space.

## I    PROMPT USED IN EXPERIMENT

The prompt used for slot filling task is shown as below, in which the input reference transcription refers to the reference transcription as the input. For example, with input "remind me about my business meeting at 3 and 45 pm", the expected output is '{"time": "3 and 45 pm"}'

## J    LICENSES FOR EXISTING ASSETS

Models and datasets we used in this work were all downloaded from HuggingFace website, expect SLURP which is downloaded from `https://github.com/pswietojanski/slurp`. The licenses and paths for each asset used is listed below:

We provide the following links to special licenses below:

- Modified MIT License for GPT2-Large: `https://github.com/openai/gpt-2/blob/master/LICENSE`

USER: Consider the following list of slot types provided to you:
"event_name", "date", "person", "time", "news_topic", "relation", "list_name", "media_type",
"business_name", "weather_descriptor", "music_genre", "house_place", "game_name", "food_type",
"timeofday", "place_name", "definition_word", "email_address", "transport_agency", "movie_name",
"artist_name", "transport_type", "joke_type", "movie_type", "time_zone", "music_descriptor",
"device_type", "color_type", "meal_type", "player_setting", "podcast_name", "email_folder",
"song_name", "change_amount", "business_type", "personal_info", "radio_name", "coffee_type",
"audiobook_author", "audiobook_name", "currency_name", "playlist_name", "podcast_descriptor",
"general_frequency", "music_album", "app_name", "order_type", "transport_name",
"transport_descriptor", "cooking_type", "ingredient", "alarm_type", "drink_type", "sport_type",
"game_type"
Now consider the following sentence(s) containing one or more of the above slot types. Can you extract
slots belonging to that slot list and their values in json format i.e. {"slot type": "value"}? ONLY print out
the json, or only print {} if no slot.
"{input reference transcription}"
ASSISTANT:

| Model/Dataset | License | Huggingface Path |
|---|---|---|
| GPT2-Large | Modified MIT License | openai-community/gpt2-large |
| OPT-1.3B | MIT license | facebook/opt-1.3b |
| Pythia-1.4B | Apache 2.0 | EleutherAI/pythia-1.4b |
| BLOOM-1B1 | RAIL License v1.0 | bigscience/bloom-1b1 |
| TinyLlama_v1.1 | Apache 2.0 | TinyLlama/TinyLlama_v1.1 |
| Qwen-7B | Tongyi Qianwen LICENSE AGREEMENT | Qwen/Qwen-7B |
| Llama2-7B | Custom commercial license | meta-llama/Llama-2-7b-hf |
| SciQ | CC BY-NC 3.0 DEED | allenai/sciq |
| SLURP | CC BY 4.0 | N/A |
| CosmosQA | CC BY 4.0 | allenai/cosmos_qa |

- RAIL License v1.0 for BLOOM-1B1: `https://huggingface.co/spaces/bigscience/license`

- Tongyi Qianwen LICENSE AGREEMENT: `https://github.com/QwenLM/Qwen/blob/main/Tongyi%20Qianwen%20LICENSE%20AGREEMENT`

- Custom commercial license for Llama-2: `https://ai.meta.com/resources/models-and-libraries/llama-downloads`

