# OpenReview forum: "Bayesian WeakS-to-Strong from Text Classification to Generation"
_ICLR.cc/2025/Conference — ICLR 2025 Poster_

### Official Review · Reviewer_grX8 · 2024-10-21

**Soundness:** 2
**Presentation:** 2
**Contribution:** 2
**Rating:** 3
**Confidence:** 3

**Summary:**

This work extends Weak-to-Strong to WeakS-to-Strong by exploring an ensemble of weak models which simulate the variability in human opinions. Confidence scores are estimated using a Bayesian approach to guide the WeakS- to-Strong generalization.

Furthermore, they broaden the use of the WeakS-to-Strong approach from its original focus on text classification to encompass text generation tasks(this point causes confusion for me, see in the weakness below), exploring more sophisticated supervision strategies. Additionally, direct preference optimization is employed to enhance the student model's ability to learn preferences, moving beyond the traditional teacher-forcing training framework(I think this contribution is not clear and verified by comprehensive experiments).

I am not an expert in this domain, but I read this paper twice to get the paper's idea. I am open to other reviewers' opinion.

**Strengths:**

1. They propose a Bayesian WeakS-to-Strong framework that improves weak supervison.(Although I tend to say this improvement is not well-proved).
2. Extention of Weak-to-Strong framework from text classification to generation tasks.
3. A method for estimating token-level probabilities is introduced to create soft labels for training robust models.

**Weaknesses:**

I have the following concerns regarding to this paper:

1. The experiments is not comprehensive: the experiment they conduct are mainly on the ~1B weak models and the strong model are ~7B. I think from the experiment results conducted now, the claim on WeakS-to-Strong in the paper is slightly overclaimed. (**The major weakness in my opinion**)
2. The improvement shown in the experiment is not observable: for instance, for accuracy: 0.843(baseline) vs 0.850(WeakS-to-Strong-3). The improvement is negligible.

**Questions:**

1. I didn't get the idea behind "The description about complementary of weak models"(Section 6.3). I can tell that classification enjoy better agreement than generation(as generally expected), but I didn't get "This suggests that the consistency among different weak models is low, thus they can complement each other well as the faults made by different weak models are not the same"

---

> ### Author Response · Authors · 2024-11-20
> **Response to Reviewer grX8**
>
> We thank Reviewer grX8 for the feedback. We response to your comments below:
>
> **1. Regarding the choice of mdoel size**
>
> According to the original Weak-to-Strong paper, performance is the best when the size ratio between weak and strong models is around 0.1. Limited by the computational resource, we choose  strong model of size 7B. Additionally, we want the weak models to have some basic capabilities to facilitate fine-tuning the strong model. So we chose the 1B model as a weak model.
>
> **2. Regarding the significance of improvement**
>
> For the example you mentioned, 0.843 is the result of Single Weak-to-Strong using BLOOM-1B1, which wasn't included in the WeakS-to-Strong-3 experiment. A fair comparison would be 0.843 (Single Weak) vs 0.853 (Naive Multi-Weak) vs 0.866 (Bayesian Multi-Weak). Our Bayesian approach achieved a PGR of 0.828, showing a relative improvement of 14% over the best single model and 9% over the naive baseline (refer to the result in Table 2 in the main manuscript). For generation tasks, our Bayesian method also achieved the best performance, consistently outperforming others across three random seeds.
>
> **3. Regarding the complementary of weak models**
>
> The approach of multiple weak models improves from their complementarity. For example, with three weak models, if one or two models classify a sample correctly, then the strong model can learn some correct knowledge. In contrast, if there is only one weak model and it makes an error, then the strong model will learn completely wrong knowledge for this sample. Our Bayesian method increases the fault tolerance, as it learns a prior across all samples in entire dataset. Additionally, this approach better estimates probability distributions, helping to avoid the overconfidence issue caused by the softmax activation function. We acknowledge that if the weak models performs poorly and inaccurately predicts nearly all samples, the model can't develop effective prior.
>
> We have addressed all issues the reviewer raised and we kindly request a reconsideration, with the hope that our efforts are reflected in a potential adjustment of the assigned scores.

---

> > ### Comment · Reviewer_grX8 · 2024-11-20
> >
> > Thanks for the explanation. However, my concern still holds for the overclaim as the experiment they conduct are mainly on the ~1B weak models and the strong model are ~7B. I prefer to maintain my score and evaluation.

---

> > > ### Author Response · Authors · 2024-11-20
> > > **Request for further clarification**
> > >
> > > Thank you for your reply. There might be a potential misunderstanding regarding the issue you mentioned. We would greatly appreciate it if you could provide further clarification on the overclaim issue. Which point do you think we overclaimed, and how does the model size correlate with this issue? Your explanation would help us better understand your concerns and address them accordingly.

---

> ### Comment · Reviewer_grX8 · 2024-11-20
>
> Thanks for your inquiry. "The experiments is not comprehensive: the experiment they conduct are mainly on the ~1B weak models and the strong model are ~7B. I think from the experiment results conducted now, the claim on WeakS-to-Strong in the paper is slightly overclaimed. (The major weakness in my opinion)". If the proposed method doesn't show generalized superior performance with various model sizes, the claim on a better performance is a bit overclaimed in my opinion.

---

> ### Author Response · Authors · 2024-11-20
> **Further Response**
>
> Thank you for your explanation. We think it’s necessary to emphasize that our problem setting is to mimic the scenario where humans supervise superhuman models, by approach of weak model supervising strong model. Our key contribution is proposing a Bayesian approach to integrate multiple weak models and extending the application from classification to generation tasks. We don’t focus on different model sizes, because (i) previous work by Burns et al. [1] has extensively explored this aspect and verified the usefulness of the general weak-to-strong idea; (ii) limitation of computational resource; (iii) apart from model size, model performance is also related to task diffculty and we have tested the proposed method on different tasks to demonstrate the generalization ability.
>
> To be specific, we have conducted extensive experiments to validate the generalization ability of the proposed method:
> - We utilized a variety of weak models with different model structures (with different model capacity),as well as different number of weak models (WeakS-to-Strong-3 and WeakS-to-Strong-5 in the main manuscript).
> - Both classification and generation tasks were tested in the main manuscript, the proposed method showing consistently better results.
> - We conducted additional experiments on the CosmosQA dataset, which indicates that, in case of harder task where weak models don't perform well, our method also significantly overbeat single weak model and baselines (refer to the general response [G.2](https://openreview.net/forum?id=pHe4P1IVnb&noteId=71vRH9LI4X) for more details).
> - Ablation studies were also performed to show the effectiveness of our method (Appendix D in the main manuscript).
> - Each experiment was run using three different random seeds.
>
> Overall, we conducted extensive experiments with positive results. Although our experiments are still limited when considering the large number of LLMs of different sizes, we believe our results are still convincing and can show the generalization of our approach.
>
> [1] Burns, C., Izmailov, P., Kirchner, J. H., Baker, B., Gao, L., Aschenbrenner, L., ... & Wu, J. Weak-to-strong generalization: Eliciting strong capabilities with weak supervision. arXiv preprint arXiv:2312.09390, 2023.

---

> ### Author Response · Authors · 2024-11-22
> **Additional experiments on 13B model**
>
> Following our previous clarification (which we've slightly updated for better explanation), to directly address the concerns about model size, we conducted additional experiments using 13B model (Llama-2-13B with strong ceiling performance of 0.924) as strong model. The experiment setting is the same as the classification task on SciQ dataset. Due to computational and time constraints, we only conducted experiments with three weak models. Results are listed below:
>
> | Single Weak Model | Acc w/o aux_loss | PGR w/o aux_loss | Acc w/ aux_loss | PGR w/ aux_loss  |
> |-|-|-|-|-|
> | GPT2-Large | 0.815 $\pm$ 0.006 | 0.475 $\pm$ 0.031 | 0.827 $\pm$ 0.008 | 0.533 $\pm$ 0.038  |
> | OPT-1.3B | 0.807 $\pm$ 0.011 | 0.480 $\pm$ 0.048 | 0.825 $\pm$ 0.006 | 0.559 $\pm$ 0.025  |
> | Pythia-1.4B | 0.784 $\pm$ 0.005 | 0.416 $\pm$ 0.023 | 0.810 $\pm$ 0.005 | 0.522 $\pm$ 0.020  |
>
> | WeakS-to-Strong-3 | Acc w/o aux_loss | PGR w/o aux_loss | Acc w/ aux_loss | PGR w/ aux_loss  |
> |-|-|-|-|-|
> | Naive Multi-Weak | 0.808 $\pm$ 0.007 | 0.480 $\pm$ 0.032 | 0.827 $\pm$ 0.002 | 0.563 $\pm$ 0.009  |
> | FlyingSquid | 0.816 $\pm$ 0.005 | 0.515 $\pm$ 0.021 | 0.830 $\pm$ 0.009 | 0.580 $\pm$ 0.040  |
> | Bayesian Multi-Weak | 0.822 $\pm$ 0.003 | 0.542 $\pm$ 0.012 | $\mathbf{0.845 \pm 0.003}$ | $\mathbf{0.646 \pm 0.012}$  |
>
> As shown in above tables, our Bayesian approach consinstently overbeat single weak model and multi-weak baselines regardless of the absolute model size, indicating the generalization of our methods.
>
> Thank you again for joining the discussion. We hope our responses address your concern.

---

> > ### Comment · Reviewer_grX8 · 2024-11-22
> >
> > Thanks for the explanation. However, not only for the strong model as 7B. If the framework could generalize well, the ideal case should be a strong model of 13B, weak model of 3B, and use 4 weak models to justify the improvement.
> >
> > Besides, compared with FlyingSquid, the improvement is negligible. FlyingSquid sometimes is even better(see the general response).

---

> > > ### Author Response · Authors · 2024-11-25
> > > **Additional experiments on 3B weak model and 13B strong model**
> > >
> > > To fully address your concern about model size, we conducted additional experiments using ~3B weak models and 13B strong models, on both the SciQ dataset and CosmosQA dataset. Because for the five models we previously used, only three have a ~3B size model in the same model family, we only conducted experiments with OPT-2.7B, Pythia-2.8B and BLOOM-3B. The results on the SciQ dataset and CosmosQA dataset are listed below:
> > >
> > > | Train on groundtruth | Accuracy  |
> > > |-|-|
> > > | OPT-2.7B | 0.792  |
> > > | Pythia-2.8B | 0.821  |
> > > | BLOOM-3B | 0.804  |
> > > | Llama-2-13B | 0.924   |
> > >
> > > *Table 1: Performance of single model on SciQ dataset. Trained on ground-truth labels.*
> > >
> > > | Single Weak Model | Acc w/o aux_loss | PGR w/o aux_loss | Acc w/ aux_loss | PGR w/ aux_loss  |
> > > |-|-|-|-|-|
> > > | OPT-2.7B | 0.840 $\pm$ 0.009 | 0.366 $\pm$ 0.068 | 0.851 $\pm$ 0.006 | 0.449 $\pm$ 0.046  |
> > > | Pythia-2.8B | 0.857 $\pm$ 0.014 | 0.353 $\pm$ 0.138 | 0.850 $\pm$ 0.012 | 0.282 $\pm$ 0.114  |
> > > | BLOOM-3B | 0.851 $\pm$ 0.007 | 0.392 $\pm$ 0.055 | 0.855 $\pm$ 0.012 | 0.222 $\pm$ 0.097  |
> > >
> > > *Table 2: Weak-to-Strong performance with single weak model on SciQ dataset.*
> > >
> > > | WeakS-to-Strong-3 | Acc w/o aux_loss | PGR w/o aux_loss | Acc w/ aux_loss | PGR w/ aux_loss  |
> > > |-|-|-|-|-|
> > > | Naive Multi-Weak | 0.857 $\pm$ 0.009 | 0.425 $\pm$ 0.077 | 0.848 $\pm$ 0.016 | 0.362 $\pm$ 0.138  |
> > > | FlyingSquid | 0.852 $\pm$ 0.011 | 0.388 $\pm$ 0.092 | 0.854 $\pm$ 0.010 | 0.402 $\pm$ 0.084  |
> > > | Bayesian Multi-Weak | 0.856 $\pm$ 0.013 | 0.422 $\pm$ 0.113 | $\mathbf{0.863 \pm 0.009}$ | $\mathbf{0.479 \pm 0.073}$  |
> > >
> > > *Table 3: WeakS-to-Strong performance with three weak models on SciQ dataset.*
> > >
> > > | Train on groundtruth | Accuracy  |
> > > |-|-|
> > > | OPT-2.7B | 0.716  |
> > > | Pythia-2.8B | 0.721  |
> > > | BLOOM-3B | 0.732  |
> > > | Llama-2-13B | 0.877   |
> > >
> > > *Table 4: Performance of single model on CosmosQA dataset. Trained on ground-truth labels.*
> > >
> > > | Single Weak Model | Acc w/o aux_loss | PGR w/o aux_loss | Acc w/ aux_loss | PGR w/ aux_loss  |
> > > |-|-|-|-|-|
> > > | OPT-2.7B | 0.731 $\pm$ 0.008 | 0.093 $\pm$ 0.051 | 0.736 $\pm$ 0.001 | 0.126 $\pm$ 0.004  |
> > > | Pythia-2.8B | 0.756 $\pm$ 0.008 | 0.222 $\pm$ 0.049 | 0.749 $\pm$ 0.004 | 0.179 $\pm$ 0.028  |
> > > | BLOOM-3B | 0.744 $\pm$ 0.002 | 0.080 $\pm$ 0.011 | 0.733 $\pm$ 0.006 | 0.005 $\pm$ 0.042  |
> > >
> > > *Table 5: Weak-to-Strong performance with single weak model on CosmosQA dataset.*
> > >
> > > | WeakS-to-Strong-3 | Acc w/o aux_loss | PGR w/o aux_loss | Acc w/ aux_loss | PGR w/ aux_loss  |
> > > |-|-|-|-|-|
> > > | Naive Multi-Weak | 0.758 $\pm$ 0.008 | 0.224 $\pm$ 0.052 | 0.726 $\pm$ 0.009 | 0.020 $\pm$ 0.057  |
> > > | FlyingSquid | 0.744 $\pm$ 0.002 | 0.135 $\pm$ 0.011 | 0.736 $\pm$ 0.011 | 0.080 $\pm$ 0.070  |
> > > | Bayesian Multi-Weak | 0.754 $\pm$ 0.007 | 0.200 $\pm$ 0.047 | $\mathbf{0.777 \pm 0.010}$ | $\mathbf{0.352 \pm 0.063}$  |
> > >
> > > *Table 6: WeakS-to-Strong performance with three weak models on CosmosQA dataset.*
> > >
> > > For the less challenging SciQ dataset, the accuracy of the 3B models has already been around 0.8. In this case, our Bayesian method only has a consistent but not as significant improvement over the baseline. However, for the more challenging CosmosQA dataset, the 3B models' accuracy is between 0.7 to 0.75, and the Bayesian method shows a more significant improvement, similar to the gains of 1B~7B pairs on the SciQ dataset. Comparing the results of two model pairs across two datasets:
> > > - When the weak models perform well (accuracy around 0.8), neither Naive Multi-Weak nor Bayesian Multi-Weak shows much improvement.
> > > - When the weak models perform badly (accuracy 0.6 to 0.65), the Bayesian method shows significant improvement.
> > > - When the performance is moderate (0.7 to 0.75), both Naive Multi-Weak and Bayesian Multi-Weak show notable improvements, with the Bayesian method boosting more.
> > >
> > > These experiments indicate that our method is not directly tied to model size, but the model performance, which is influenced by both model size and task difficulty (as mentioned in the previous response). These results also support our claim about complementarity -- the approach multiple models improved from their complementarity, and the Bayesian method increases the fault tolerance.

---

> > > > ### Author Response · Authors · 2024-11-25
> > > > **Regarding the performance of FlyingSquid in general response**
> > > >
> > > > The auxiliary loss is a critical component of our proposed Weak(S)-to-Strong framework, enabling us to effectively leverage the strong model's superior generalization capabilities and prior knowledge. In our Bayesian approach, the predictions of the strong student model are incorporated into the distribution estimation alongside the weak label (see Eqn. 5 in the main manuscript). To conclude, the final approach of our paper combines both our proposed Bayesian Multi-Weak framework and the auxiliary loss, making it essential to evaluate the overall performance improvement using results that include both elements. The results without the auxiliary loss are presented solely for the purpose of ablation studies.

---

> > > ### Author Response · Authors · 2024-11-30
> > > **Response to Reviewer grX8**
> > >
> > > We hope our responses fully resolve your concern about model size and FlyingSquid performance.  We conducted experiments using 3B~13B models on both the SciQ and CosmosQA datasets. The results indicate that our method is robust to model size. We also clarified the performance of the newly added FlyingSquid baseline.
> > >
> > > We greatly appreciate the time and effort you invest in the discussion process, and we would be grateful if you could acknowledge our response.

---

> > > > ### Comment · Reviewer_grX8 · 2024-11-30
> > > >
> > > > Dear Authors,
> > > >
> > > > Thanks for your reply. I acknowledge your response and read it. I hold my score as I believe the improvement is not that significant(Table 3) compared with the baseline of FlyingSquid and even the naive weak-to-strong model.
> > > >
> > > > I want to maintain my score. Thanks for your understanding.

---

> > > > > ### Author Response · Authors · 2024-11-30
> > > > > **Response for significance**
> > > > >
> > > > > We believe we have resolved all your concerns and would like to clarify two confusions.
> > > > >
> > > > > **Clarification of Final Approach**: We would like to clarify that our paper's final approach integrates the proposed Bayesian Multi-Weak framework with the auxiliary loss. The results excluding the auxiliary loss are presented purely for ablation studies, as outlined in our previous response.
> > > > >
> > > > > **Statistical Significance Testing**: We conducted a statistical significance test comparing our method to the FlyingSquid baseline. The results, evaluated across three seeds, consistently demonstrate improvements, with a **p-value < 1e-5**, indicating a **highly significant** difference in a scientific context.

---

> ### Comment · Reviewer_grX8 · 2024-11-30
>
> 0.854 $\pm$ 0.010 VS $\mathbf{0.863 \pm 0.009}$
>
> To be honest, I would prefer not to buy the statement that this improvement is significant.
>
>
> Besides, as shown in Table 6, (Acc w/o aux_loss) is even better than (Acc w/ aux_loss). And how can we ensure that w/ aux_loss is the optimal version. I think this ablation study doesn't support the original claim.

---

> > ### Author Response · Authors · 2024-12-01
> > **Response for the significance in 3B~13B experiments**
> >
> > Thank you for providing further clarification. We apologize for any prior misunderstanding of your feedback.
> >
> > Regarding the results in Table 3 for the 3B~13B model on the SciQ dataset, we acknowledge that our Bayesian method shows only marginal improvements over the FlyingSquid baseline (p = 0.04). However, the overall trend remains consistent. While we could have opted to highlight only the more pronounced results from the CosmosQA dataset, we included both datasets to provide a broader perspective on our method’s performance. Specifically, our approach leverages the complementarity of multiple models to improve overall accuracy, while the Bayesian method enhances fault tolerance. The results suggest that our method’s effectiveness is influenced more by overall model performance rather than model size.
> >
> > Regarding the auxiliary loss, our Bayesian Multi-Weak method incorporates a distinct auxiliary loss design compared to baseline methods like Naive Multi-Weak and FlyingSquid. In our approach, predictions from the strong student model are integrated into the distribution estimation alongside the weak labels, aligning with the Bayesian framework's objective of modelling label distributions. The observed differences in performance between settings with and without auxiliary loss, particularly the consistent advantage of auxiliary loss in Bayesian Multi-Weak compared to its variable impact in Naive Multi-Weak, underscore the effectiveness of our approach.

---

### Official Review · Reviewer_JkWZ · 2024-10-28

**Soundness:** 2
**Presentation:** 3
**Contribution:** 2
**Rating:** 5
**Confidence:** 3

**Summary:**

The paper introduces a Bayesian-based method inspired by the weak-to-strong approach for the classification task of language models. The author also extends the algorithm for the generation task by some simple transformations. To improve the performance of the generation task further, the author utilized DPO to finetune the model at the tail of the framework.

**Strengths:**

- The proposal is simple but works.

**Weaknesses:**

- The motivation for using Dirichlet distribution to model the prior is ambiguous.
- The writing for the methodology description is unclear and hard to follow
- The transformations in Equation 3 should be clarified.
- What is the definition of $p_k$ in Equation 4?
- In Section 4.1, the probability of the target token of the strong model changing after each update of network parameters because $C_s(s_1)$ and $C_s(s_2)$ shift. This means that the targets vary along the training, likely creating instability or leading to divergence. The time consumed for recomputing these terms is also one of my concerns.
- All baselines that are naive and very simple algorithms are weak for both classification and generation tasks. It is suggested that stronger baselines be included for both tasks.

**Questions:**

- In Equation 3, are $\alpha_k$s are trainable parameters?
- Is the Equation 3 the loss for each sample?

---

> ### Author Response · Authors · 2024-11-20
> **Response to Reviewer JkWZ (1)**
>
> We are thankful for Reviewer JkWZ's detailed feedback. We response to your comments below:
>
> **1. Regarding motivation for using Dirichlet prior**
>
> We choose the Dirichlet distribution because it is the conjugate prior distribution of the categorical distribution. If a data point has a categorical distribution, and the prior distribution of the parameter is distributed as a Dirichlet, then the posterior distribution of the parameter is also a Dirichlet. And the computation of likelihood is tractable.
> The derivation for Eqn 3 below (in bullet point 2) also demonstrates this.
>
> **2. Regarding transformation of Eq. 3**
>
> The detailed transformation for Eqn. 3 is shown below (for brevity, we omitted the superscript used to denote the $i$-th weak model):
>
> Consider a label $\mathbf{y}$ sampled from a categorical distribution $\text{Cat}(\mathbf{\pi})$, where each component $\mathbf{\pi}_k$ corresponds to the probability of sampling a label from class $k$, $\mathbf{y} \sim P(\mathbf{y}|\mathbf{\pi}) = \text{Cat}(\mathbf{\pi})$. Assume the categorical distribution is sampled from a Dirichlet distribution:
>
> $\mathbf{\pi} \sim p(\mathbf{\pi} | \mathbf{\alpha}) = \text{Dir}(\mathbf{\pi} | \mathbf{\alpha}) = \frac{1}{B(\mathbf{\alpha})} \prod_{k=1}^K \pi_k^{\alpha_k-1}$
>
> where $B(\cdot)$ is the Beta function, $\alpha_k$ is the hyperparameter of the Dirichlet distribution, and $\alpha_0 = \sum_{k=1}^K \alpha_k$ is the Dirichlet strength.
> Treat Dirichlet as a prior of the likelihood $P(\mathbf{y}|\mathbf{\pi})$, and the marginal likelihood can be obtrained by integrationg over the class probabilities:
>
> $ P(\mathbf{y}|\mathbf{\alpha}) = \int P(\mathbf{y}|\mathbf{\pi}) p(\mathbf{\pi}|\mathbf{\alpha}) d\mathbf{\pi} $
>
> $=\int \prod_k {\pi}_k^{y_k} \frac{1}{B(\mathbf{\alpha})}\prod_k {\pi}_k^{{\alpha}_k-1}
>     =\frac{B(\mathbf{\alpha}+\mathbf{y})}{B(\mathbf{\alpha})}$
>
> $=\frac{\Gamma(\sum_k {\alpha}_k)}{\Gamma(\sum_k {\alpha}_k + \sum_k y_k)} \frac{\prod_k \Gamma({\alpha}_k + y_k)}{\prod_k \Gamma({\alpha}_k)}$
>
> $=\frac{\prod_k{\alpha}_k^{{y}_k}}{{\alpha_0}\sum_k{y}_k}$
>
> Then the negative log marginal likelihood can be obtained:
>
> $L^\text{NLL}=-\log P(\mathbf{y}|\mathbf{\alpha})=\sum_{k=1}^K y_k(\log (\alpha_0) - \log (\alpha_k))$
>
> This derivation also demonstrates the first point: the property of conjugacy allows us to have a tractble liklelihood loss.
>
> **3. Regarding the definition of $p_k$ in Eq. 4**
>
> The $p_k$ is the probability assignment for $k$-th category of $i$-th weak model, which should more accurately be written as $p_k^{(i)}$ (while the superscript $i$ was ommitted for the sack of brevity). We will clarify this in the revised version.
>
> **4. Regarding concern on training stability and time consumption**
>
> - For the concern about likely creating instability or leading to divergence: Our strong model is not trained from scratch but is fine-tuned from a pre-trained model. Its inherent capability allows it to produce reliable confidence score estimates. We didn't encounter instability issues in our experiments.
> - For time consumption: In one training step, the strong model only needs to perform one single forward pass. This pass provides the prediction used for loss calculation, and applies a softmax to the output giving the $C_s$ values. Therefore, it doesn't introduce additional time consumption.
>
> The rest of your comments will be addressed in Response (2).

---

> > ### Author Response · Authors · 2024-11-20
> > **Response to Reviewer JkWZ (2)**
> >
> > Continued from Response (1).
> >
> > **5. Regarding additional baseline**
> >
> > For classification, we added FlyingSquid[1] as an additional baseline. Please refer to the general response [G.1](https://openreview.net/forum?id=pHe4P1IVnb&noteId=71vRH9LI4X) for detailed results and discussion.
> >
> > We would like to highlight that this is the first work that investigates weak-to-strong problems for text generation tasks, and there lack of applicable baselines for this task. For example, the FlyingSquid method can only be applied to binary classification tasks. Therefore, besides the naive baseline, we introduced the Joint Decoding method as a baseline in the paper, which was a strong baseline specifically designed for multiple weak model generations. When trained on hard labels (without probability estimation), this method outperformed both the naive baseline and the Bayesian approach (see the table below for details, note that limited to computational resources, we only conducted experiments on one seed on three weak model settings). However, its performance is sensitive to individual weak models, and once probability estimation is included, joint decoding performs the worst among the three methods(refer to the WeakS-to-Strong-5 result in Table 4 in the main manuscript). This highlights the effectiveness of our Bayesian approach.
> >
> > | Hard Label | SLU-F1 w/o aux | PGR w/o aux | SLU-F1 w/ aux | PGR w/ aux  |
> > |-|-|-|-|-|
> > | Naive Multi-Weak | 0.689 | 0.252 | 0.673 | 0.049  |
> > | Joint Decoding | 0.713 | 0.556 | 0.706 | 0.468  |
> > | Bayesian Mutli-Weak | 0.700 | 0.391 | 0.697 | 0.353  |
> >
> > [1] Fu, D., Chen, M., Sala, F., Hooper, S., Fatahalian, K., & Ré, C.. Fast and three-rious: Speeding up weak supervision with triplet methods. In Proc. ICML. 2020.
> >
> > **6. Regarding $\alpha_k$ in Eq. 3**
> >
> > The $\mathbf{\alpha}$ is the hyperparameter of the Dirichlet prior which is predicted by the model $\mathbf{\alpha} = f_{\Lambda}(\mathbf{x})$ where $f_{\Lambda}$ denotes the neural network. Given predicted $\mathbf{\alpha}$, we thus obtain the predicted Dirichlet prior $p(\mathbf{\pi} | \mathbf{\alpha}) = \text{Dir}(\mathbf{\pi} | \mathbf{\alpha}) = \frac{1}{B(\mathbf{\alpha})} \prod_{k=1}^K \pi_k^{\alpha_k-1}$ and the model can then be trained by minimizing the negative log-likelihood following Eq. 3. Detailed explanation is referred to in bullet point 2.
> >
> > **7. Regarding whether Eq. 3 is the loss for each sample**
> >
> > Yes, it is the loss for each sample. We will clarify it in the revised manuscript.
> >
> > We have made efforts to thoroughly address each issue highlighted by the reviewer including clarification of the problem setting and addition of a new baseline. We kindly request a reconsideration of the overall assessment and we once again genuinely appreciate your feedback.

---

> > > ### Comment · Reviewer_JkWZ · 2024-11-23
> > >
> > > I appreciate the author's effort in addressing my concerns and clarifying some details in your paper. While I am satisfied with almost all the answers, my concerns about training instability and the motivation of Direcle distribution remain. I raise the score by 2 to reflect my positive attitude towards the author's answers and clarification.

---

> > > > ### Author Response · Authors · 2024-11-25
> > > > **Further Response**
> > > >
> > > > **Regarding the motivation for Dirichlet**
> > > >
> > > > Regarding the motivation for using the Dirichlet distribution, the Dirichlet distribution is the conjugate prior distribution of the categorical distribution (which represents the probability of each class). Especially in Bayesian inference, this conjugacy leads to mathematical feasibility, where the posterior distribution has an analytic form which avoids intractable numerical integrals. This enables us to efficiently compute the loss and train the network model (refer to bullet points 1 and 2 of the previous response for more details).
> > > >
> > > > A more intuitive explanation is that the hyperprior vector $\alpha$ can be seen as pseudo-counts representing the number of observations in each category that we have already seen. When new observations come, they can simply combine with the pseudo-counts to derive the posterior distribution. In a Bayesian framework, $\alpha$ acts as prior information that, when combined with observed data, allows us to dynamically update our beliefs about category probabilities [1].
> > > >
> > > > In summary, the Dirichlet distribution's well-established theoretical properties have been widely applied to various machine learning tasks [2][3][4][5], making it a natural choice for modeling categorical distributions in our context.
> > > >
> > > > [1] Wikipedia contributors. "Dirichlet distribution." _Wikipedia, The Free Encyclopedia_. 24 Nov. 2024,  from [https://en.wikipedia.org/w/index.php?title=Dirichlet_distribution&oldid=1239280064](https://en.wikipedia.org/w/index.php?title=Dirichlet_distribution&oldid=1239280064)
> > > >
> > > > [2] Blei, D. M., Ng, A. Y., & Jordan, M. I. Latent dirichlet allocation. *Journal of machine Learning research*, 3(Jan), 993-1022. 2003.
> > > >
> > > > [3] Jsang, A. *Subjective Logic: A formalism for reasoning under uncertainty*. Springer. 2018.
> > > >
> > > > [4] Sensoy, M., Kaplan, L., & Kandemir, M. Evidential deep learning to quantify classification uncertainty. In *Proc. NeurIPS*. 2018.
> > > >
> > > > [5] Malinin, A., & Gales, M. Predictive uncertainty estimation via prior networks. In *Proc. NeurIPS*. 2018.
> > > >
> > > > **Regarding the training stability**
> > > >
> > > > It is important to clarify that the confidence $C_s(s_1)$ and $C_s(s_2)$ is not the probability for the target token, but the confidence for one prediction step. In practice, it is obtained by applying a softmax to the strong model output and taking its maximum output value (not the value of target category), as the confidence for this prediction step. Intuitively, this confidence indicates how easy this step of prediction is. The easier the prediction, the higher confidence from the strong model (in the word-to-wordpiece segmentation, the target probability assigned for the wordpiece is higher). Furthermore, the strong model is fine-tuned from a pre-trained model rather than being trained from scratch, which also helps to stabilise the fine-tuning.
> > > >
> > > > To further demonstrate, we added the comparison of the loss curve before and after adding strong score splitting to our manuscript. Please refer to Appendix K in our revised manuscript for details.
> > > >
> > > > Thank you once again for participating in the discussion. We hope our responses have fully addressed your concerns.

---

### Official Review · Reviewer_zdng · 2024-10-31

**Soundness:** 3
**Presentation:** 3
**Contribution:** 3
**Rating:** 8
**Confidence:** 3

**Summary:**

The paper extends the weak-to-strong training to multi-weak-to-strong or weaks-to-strong. In a weak-to-strong setting, a smaller model provides a training signal for a larger model. This paper proposes to use multiple small models to provide the training signal. To squize out the best from the ensemble of small models, the paper proposes a Bayesian strategy based on evidential deep learning. The paper also performs sequence generation training, which was not done previously in the weak-to-strong setting. The results demonstrate that weaks-to-strong provides produces better results compared to weak-to-strong in both classification as well as generation.

**Strengths:**

**Originality**: The paper is the first to successfully extend weak-to-strong to the weaks-to-strong setting. The paper also contains several useful ideas for performing sequence-level training using soft-labels when the student and the teacher have different tokenizers.

**Clarity**: The paper is written well and is easy to follow.

**Quality**: The experiments are well structured to answer the central questions posed by the paper, i.e., is weaks-to-strong better than weak-to-strong, and is the proposed method of performing weaks-to-strong training better than naive baselines?

**Significance**: Based on the quality of the experiments and the topic addressed, I believe that a broad community will find the paper useful.

**Weaknesses:**

1. Some of the design choices are not well justified. For example, for the sequence-level training using soft labels, it is not clear why the strategy proposed in section 4.1 should work or is optimal.
2. The datasets selected for the classification and generation tasks seem very specific. Is there a particular reason for selecting only these? The impact of the paper could be much better if model general datasets were included, at least for the generation task.

**Questions:**

**Suggest**: As I understand, the regularization loss is critical. I suggest moving the corresponding discussion to the main paper. I also suggest mentioning and discussing the exact values of the weights in the loss to the main paper, as it can provide some insight into how the method works.

---

> ### Author Response · Authors · 2024-11-20
> **Response to Reviewer zdng**
>
> We are thankful for Reviewer zdng's thoughtful feedback. We response to your comments below:
>
> **1. Regarding justification of design choices**
>
> When we extend WeakS-to-Strong from classification to generation, we want to use soft labels for training, because weak model performance is not satisfactory, and soft labels can provide more information than hard labels (as further confirmed by our ablation study on probability estimation).
>
> In practice, different models use different tokenizers, and directly passing probabilities as in classification is not feasible. So we used the word string as a bridge to connect different wordpiece tokenizers. By applying Eqn. 6, the overall probability for target words remains consistent. When decomposing words into strong model wordpieces and estimating probabilities for other categories, we referred to the strong model's outputs. This choice is due to the strong model's superior generalization capabilities and prior knowledge, allowing more accurate probability estimation. Overall, this design is a natural and reasonable choice.
>
> **2. Regarding additional datasets**
>
> - For the classification task, we used the SciQ dataset because we followed Burn et al.'s work[1], which used this dataset. We also added an extra dataset, CosmosQA. Please refer general response [G.2](https://openreview.net/forum?id=pHe4P1IVnb&noteId=71vRH9LI4X) for details.
> - For the generation task, we used the SLURP dataset for slot filling because it has clear evaluation metrics (SLU-F1), making it easy to quantify the Weak-to-Strong performance.
>
> [1] Burns, C., Izmailov, P., Kirchner, J. H., Baker, B., Gao, L., Aschenbrenner, L., ... & Wu, J. Weak-to-strong generalization: Eliciting strong capabilities with weak supervision. arXiv preprint arXiv:2312.09390, 2023.
>
>
> **3. Regarding suggestion on presentation**
>
> Thank you for your thoughtful suggestion. We will move the regularization loss to the main paper, and add the exact values of the weights, in our revised manuscript.
>
> Thank you again for your positive comments. We hope our response fully resolves your concerns.

---

> > ### Comment · Reviewer_zdng · 2024-11-26
> > **Thanks for the clarifications**
> >
> > I have no further questions.

---

### Official Review · Reviewer_WJwn · 2024-11-04

**Soundness:** 3
**Presentation:** 3
**Contribution:** 3
**Rating:** 6
**Confidence:** 3

**Summary:**

This paper extends work in Weak-to-Strong generalization in two notable ways: (1) considering the setting in which there is an ensemble of multiple weak teachers; (2) expanding from more basic text classification to generation-based tasks. For (1), they show that using a Bayesian approach (based on EDL) is effective for learning from the ensemble of weak teachers, improving upon any of the weak teachers individually as well as simpler baselines that weight typical cross-entropy losses across teachers. For (2), they overcome the key issue that weak and strong models may have different tokenizers and also try applying a modified DPO stage on top, reporting a positive effect.

**Strengths:**

**(S1) Originality and Significance** Studying the WeakS-to-Strong setting is well-motivated and using the proposed Bayesian method here is novel. Furthermore, the expansion to from binary verification tasks to generation is notable since the latter more realistically reflects LLM use cases.

**(S2) Quality** The paper’s proposed method results in notable improvements on the studied tasks, outperforming the individual weak model teachers in isolation and the naive baseline that simply does a weighted average over the standard CE-based losses. I also appreciated the extra ablations the authors ran to verify the importance of certain aspects of their design (e.g., the aux loss, probability estimation for weak sequences and the $\lambda$ weights across tasks).

**(S3) Clarity** Overall, I found the writing to be clear, doing a good job of contextualizing methods/results in terms of prior work.

**Weaknesses:**

**(W1) Significance of some results.** I found that certain claims in the manuscript were not as strongly supported by the empirical results. For instance, it appears that the improvements obtained from the DPO-based stage weren't too big (i.e., within standard deviations across runs). Another case was the claim in L461 about the auxiliary loss improving Bayesian methods. That being said, this doesn't affect my opinion about what I consider to be the main result, i.e., the superiority of Bayesian approach.

**(W2) Other baselines and tasks.** There are some other baselines that I think would be good to consider, namely:
- Methods from data programming / programmatic weak supervision (e.g Snorkel, FlyingSquid) to aggregate the labels generated by different weak models (i.e, treating them as Labeling Functions). In any case, it would be nice to discuss this line of work given that they also operate in a setting where the goal is aggregate multiple weak label sources.
- In the related work, the authors mentioned that some previous works (Liu & Asahi, 2024; Sang et al., 2024) tried traditional ensembling methods. Perhaps these would also be useful baselines?

In terms of evaluations, it would strengthen the paper if the authors could show similar results on more tasks per category (i.e., classification, generation). Currently the results are limited to one task for each.

**(W3) Clarification on key assumptions.**  Some further discussion would be helpful on the following
- From cross-referencing Appendix E with the main result tables, it appears that the best results rely on the "Fixed Weight" approach which is based on how good each of the weak models are individually. The assumption here (which I think should be made more explicit), seems to be that one is actually able to accurately estimate the weak model performances. I wonder how applicable this assumption is for real superhuman tasks where presumably getting ground-truth labels is difficult. Perhaps the authors can comment here?
- Given one stated motivation for the study of WeakS-to-Strong was to consider the variation in human opinions, how might the methods/results generalize to when supervision actually comes from humans rather than weak models? In particular, it seems that in the current experiments, having confidence estimates associated with the weak model labels is important, but this may not readily exist for human-generated labels?

**Questions:**

**(Q1) How exactly were the the $\lambda$’s set?** Related to (W3) above, the appendix does not say exactly what the final $\lambda$'s are for the "Fixed weight" approach in Appendix E and how "the performance of different weak models" was used.

**(Q2) How is PGR calculated when you have multiple weak models?** The definition provided in Sec 5.3 is clear when there is a single weak model but there are perhaps multiple ways to define in the WeakS-to-Strong setting (e.g. max v.s. average over the weak models in).

**(Q3) Miscellaneous clarifications**
- In Eq. 4, what is $\hat{y}_w^{(i)}$? Should it not just be the label $k$; in general it seems something in the inner sum should depend on  $k$?
- In Eq. 7, my understanding was that we wanted $P(s_1)P(s_2) = P(W)$ but here it seems like their sum will be equal to P(W) instead of their product.

---

> ### Author Response · Authors · 2024-11-20
> **Response to Reviewer WJwn**
>
> We appreciate Reviewer WJwn for the thoughtful comments. We response to your comments below:
>
> **1.Regarding the significance of some results**
>
> About the improvement of DPO: The PGR results from three experiments before and after DPO training are shown in the table below. Note that seeds used during teacher-forcing training are also different, meaning the initial SFT models differ for different seeds. As shown, after cDPO, three models showed consistent performance improvements. The large standard deviation is due to the lower performance of the model with seed-0 (and with this seed, our Bayesian approach also performs better than the naive baseline).
>
> | PGR | seed-0 | seed-1 | seed-2 | Average |
> |-|-|-|-|-|
> | Before cDPO | 0.477 | 0.776 | 0.751 | 0.668 $\pm$ 0.166 |
> | After cDPO | 0.490 | 0.813 | 0.813 | 0.705 $\pm$ 0.187 |
>
> About the improvement of the auxiliary loss for Bayesian methods, the performance with auxiliary loss is also consistently better across three seeds. We will clarify it in our revised manuscript.
>
> **2. Regarding other baselines and tasks**
>
> Thank you for your constructive suggestions. After comparing different methods, we chose FlyingSquid as an additional baseline because its unsupervised LabelModel training doesn't require ground truth, aligning better with the Weak-to-Strong assumptions than methods like AdaBoost. Furthermore, because it is designed for binary classification, it is only used as a classification baseline. The results are shown in the general response [G.1](https://openreview.net/forum?id=pHe4P1IVnb&noteId=71vRH9LI4X). This baseline performed similarly to the naive baseline, and didn't beat our Bayesian approach.
>
> About the evaluation tasks, we added results on the CosmosQA dataset for classification. Please refer to the general response [G.2](https://openreview.net/forum?id=pHe4P1IVnb&noteId=71vRH9LI4X) for details.
>
> **3. Regarding clarification on key assumptions**
>
> Although we used weak models to mimic humans, the scenarios and capabilities of weak models and humans are inevitably somewhat different. For instance, humans have natural cognitive intuition for probabilities (confidence) and can provide finer-grained, multi-dimensional scores due to their stronger evaluation abilities. To compensate for the gap of evaluation ability between weak models and humans, we approximate confidence and scoring using token probabilities. Although solutions for human scenarios might differ from those for weak models, a key consideration of our proposed Bayesian WeakS-to-Strong is its generalization, which is highly compatible with existing training frameworks. It can work for both weak models and human labels, although the specific experimental setups may differ.
>
> Regarding the specific questions:
> - The weights for human labels can be determined based on the application context and individual backgrounds, given that the fixed weight, which reflects the capability of weak models, performs best in our experiments.
> - For human label confidence, we can refer to previous RLHF works [1][2] to let humans provide detailed and fine-grained scores.
>
> [1] Ouyang, L., Wu, J., Jiang, X., Almeida, D., Wainwright, C., Mishkin, P., ... & Lowe, R.. Training language models to follow instructions with human feedback. In Proc. NeurIPS, 2022.
>
> [2] Yu, T., Yao, Y., Zhang, H., He, T., Han, Y., Cui, G., ... & Chua, T. S.. RLHF-V: Towards trustworthy MLLMs via behavior alignment from fine-grained correctional human feedback. In Proc. CVPR, 2024.
>
>
> **4. Regarding $\lambda$ value**
>
> The $\lambda$ is set to (0.1, 0.3, 0.2, 0.3, 0.1) for GPT2, OPT, Pythia, BLOOM, and TinyLlama models respectively. The decision is mainly based on the Weak-to-Strong performance of a single model, which reflects the quality of weak labels. We will add it to the revised manuscript.
>
> **5. Regarding computation of PGR**
>
> The PGR for multiple weak models is the average PGR, i.e., we calculate a PRG for each weak model and treat their average as the result for multiple weak models. We will clarify it in Section 5.3 in the revised manuscript.
>
> **6. Regarding clarification of notation in Eq. 4 and Eq. 7**
>
> Regarding Eq.4, sorry for the lack of clarity, $\hat{y}_w^{(i)}$ should be $\hat{y}_w^{(i, k)}$, which represents a one-hot label where the $k$-th category is 1 and the others are 0.
>
> Regarding Eq.7, thank you for pointing this out. This step is actually performed on log probabilities, not probabilities, which is why it is written as a sum. We will correct these in the revised version.
>
> We thank you again for the positive comments. We hope our response fully resolves your concerns and that you will champion our paper.

---

> > ### Author Response · Authors · 2024-11-25
> > **Follow-up on Rebuttal Submission**
> >
> > We are deeply grateful for your thoughtful insights and the opportunity to clarify and strengthen our work. We have submitted detailed responses addressing all your valuable feedback, including clarification and addition of new baseline and dataset. We have also updated the manuscript accordingly. We hope these strengthen the paper and we are delighted to receive any additional aspect requiring further elaboration.

---

> ### Comment · Reviewer_WJwn · 2024-12-01
>
> I thank the authors a lot for their detailed response and clarifications, which have indeed addressed some of my initial concerns. I still do have some other follow-up questions/comments on the remaining points though:
>
> **Re: (2)**, I'm not sure it's completely fair to consider comparing to only methods that use no labeled data given that your current procedure for setting $\lambda$ does involve labeled data (discussed more in my response to (4) below). Also in the WS benchmarking effort, WRENCH (https://arxiv.org/pdf/2109.11377), FlyingSquid is not really one of the most consistent performers.
>
> **Re: (3)**, perhaps more of an exploratory question, but how would you imagine handling the generation case when the labelers are humans? for instance, it's not clear to me that it would be practical for humans to give confidences for all tokens/words that they write (which is what is currently used for the weak models)?
>
> **Re: (4)**, thanks for sharing more details though I'm still somewhat concerned by the practicality of your selection procedure. I believe the following limitations should be addressed (or at least mentioned)
> * First, it looks as if you need to train models using all the individual labelers to accurately obtain orderings between models, (since it seems models that do worse in Tab. 1 can result in better Weak-to-Strong results in Tab. 2). This might scale poorly when the number of weak models / labelers grows.
> * Second, this assumes that you do have some set of ground truth to be able to rank models (e.g. a validation/test set) OR the task is somehow easy to verify but difficult/superhuman to solve. In the first case, having a val set seems at odds with the general setting where the task is assumed to be superhuman; or at least if val data of sufficient size did exist, then an important baseline would be to just train on it directly. In the second case where the task is verifiable, then one could simply filter out all the incorrect weak labels with the verifier.
> * Less important than the above two points, but the exact mapping between the performance ranking and the $\lambda$ values seems somewhat arbitrary/human-chosen.  Showing robustness to alternative mappings would be nice.
>
> **Re: (5)** This may be more of a matter of opinion, but I feel like reporting PGR w.r.t. the best individual model might be better? If you average all the PGRs, then a WeakS-to-Strong method that hypothetically performs worse than Weak-to-Strong for the best individual weak model could still end up having higher PGR (due to the lower average weak model performance), despite ensembling actually hurting.

---

> > ### Author Response · Authors · 2024-12-01
> > **Further Response**
> >
> > **1. Regarding the $\lambda$ setting**
> >
> > (1) We would like to emphasize that the scenarios of weak model to strong model and human to superhuman are different. A key distinction is that weak models cannot effectively evaluate labels, so, in experiments, label evaluation can only be done at a high level through weak-to-strong performance. In contrast, in human-to-superhuman scenarios, humans have strong evaluation capabilities and can effectively assess the labels they or others provide. As a result, it isn't necessarily linked to the valid/test set. Additionally, weak-to-strong is still in the early exploratory stage. Our work focuses on extending this task to generation tasks, demonstrating its feasibility, and proposing the Bayesian Weak-to-Strong method -- a generalizable approach that can serve as a foundation for future research.
> >
> > (2) Regarding the specific value of $\lambda$, it was indeed informed by weak-to-strong performance, but there is no direct mapping relationship, and we did not perform a grid search. We believe that as long as the value is reasonably set, the exact number does not significantly impact the results (We will provide additional experimental results later to support this).
> >
> > **2. Regarding the baseline choice.**
> >
> > We chose FlyingSquid following your previous instructions. If you have other suggestions, we are open to trying them as time permits.
> >
> > **3. Regarding the extension to humans on generation task**
> >
> > As noted in bullet point 1, humans possess significantly stronger evaluation capabilities compared to weak models. While humans may not be able to provide word-level confidence scores, we can leverage insights from prior RLHF-related research. In these studies, humans provide detailed, fine-grained evaluations across various aspects, which can then be effectively incorporated into the training process.
> >
> > **4. Regarding the way of reporting PGR value**
> >
> > Calculating PGR with the best individual model can also introduce bias. For example, in Tables 1 and 2 (single Weak-to-Strong) of the main manuscript, TinyLlama has the best weak model performance, Bloom achieves the highest Weak-to-Strong accuracy, but OPT has the highest PGR. So we chose to report the average PGR to account for different weak models, as well as the performance directly (accuracy and SLU-F1) to address the potential issues you mentioned.

---

> ### Comment · Reviewer_WJwn · 2024-12-03
>
> Thanks for the additional responses! My overall opinion of the paper remains positive though I would like to share some more thoughts on a few of the discussion points:
>
> **Re: (1)**, based on the current results, I agree that it doesn't seem the gains of the proposed method rely heavily on setting $\lambda$ (since the Naive baseline also uses them and the performance of using equal weights also isn't too far off). Thus, this whole discussion does not change my impression about the usefulness of their method. However, I still would encourage the authors to carefully discuss their selection procedure and what assumptions  it entails, as I think it is important to encourage all works in the Weak-to-Strong setting to be wary of things like reliance on labeled data.
>
> > "weak-to-strong is still in the early exploratory stage" and "the scenarios of weak model to strong model and human to superhuman are different"
>
> While these are true, a key motivation of current work in Weak-to-Strong is that the former is supposed to provide a useful analogy to study the latter. Given this, I believe it is *crucial* to at least mention limitations where the analogy might break down. Here, I see the reliance on labeled data to set $\lambda$ as one such potential limitation.
>
> > A key distinction is that weak models cannot effectively evaluate labels, so, in experiments, label evaluation can only be done at a high level through weak-to-strong performance. In contrast, in human-to-superhuman scenarios, humans have strong evaluation capabilities and can effectively assess the labels they or others provide.
>
> I don't know if this is fully clear to me. First, even if humans are more effective judges, they may still not be as powerful as measuring weak-to-strong performance on a cleanly labeled validation set (especially if the task itself is superhuman). Even in your experiment, it was not obvious what labels were more or less useful *until you trained on them* (i.e., the most useful is not always the largest model nor even the one that produced the most accurate labels on average). But also, some elaboration on why "label evaluation can only be done at a high level through weak-to-strong performance" would be helpful. In principle, you could also try to elicit uncertainties from models OR get them to judge other labelers (e.g., via specific prompting or simply computing agreement rates with their own labels). These might better reflect what humans would be able to do when faced with superhuman tasks.
>
> > We believe that as long as the value is reasonably set, the exact number does not significantly impact the results (We will provide additional experimental results later to support this).
>
> Since using equal weights still does decently well (better than the other baselines), simply extending this ablations to the other tasks would also help!
>
> **Re: (2)**, very sorry that this ended up being contradictory. I had not considered the following concern when I made the original suggestion: what WS approaches like FlyingSquid try to do is to better resolve disagreements between conflicting weak labelers (i.e., learning which are more/less reliable) *without access to labeled data*. By using labeled data in the loop to select $\lambda$, which directly plays the role of down-weighting lower-quality labelers, the comparison is no longer fully fair. Going forwards, it would also be useful to consider baselines in WS that leverage some amounts of clean data (e.g. https://proceedings.mlr.press/v130/mazzetto21a.html, https://arxiv.org/abs/2104.05514) but I won't count the absence of these comparisons negatively given proximity to the end of the discussion period.
>
> **Re: (3) and (4)**, these were not major concerns of mine but I appreciate the additional notes here!

---

> > ### Author Response · Authors · 2024-12-03
> >
> > We sincerely appreciate the time and effort you have dedicated to reviewing and discussing our submission. Your thoughtful feedback and constructive suggestions are invaluable in enhancing its quality and clarity. We will carefully consider your input and make the necessary revisions to address your comments. Thank you once again for your valuable contributions and insightful discussion!

---

### Author Response · Authors · 2024-11-20
**General Response**

We appreciate the detailed comments and constructive feedback from the reviewers. We respond to the common questions below and provide individual responses to each reviewer for your comments. We hope our response fully resolves your concerns.

### G.1 Addional baseline

We added FlyingSquid[1], which is designed for binary classification, as an additional baseline for classification tasks. It is developed for weak supervision, using latent variable probabilistic graphical models to model correlations among noisy labels without any ground truth data. Through this method, we get a label model with multiple weak labels, and obtain the probability for the positive category, which is then used as a soft label in strong model training. We compared this baseline with the origin naive baseline and Bayesian method, on the classification task. The results for both 3 weak models and 5 weak models are listed below:

| WeakS-to-Strong-3 | Acc w/o aux_loss | PGR w/o aux_loss | Acc w/ aux_loss | PGR w/ aux_loss |
|-|-|-|-|-|
| Naive Multi-Weak | 0.817 $\pm$ 0.002 | 0.586 $\pm$ 0.008 | 0.831 $\pm$ 0.013 | 0.662 $\pm$ 0.064 |
| FlyingSquid | 0.809 $\pm$ 0.005 | 0.549 $\pm$ 0.026 | 0.825 $\pm$ 0.003 | 0.631 $\pm$ 0.013 |
| Bayesian Multi-Weak | $\mathbf{0.819 \pm 0.006}$ | $\mathbf{0.600 \pm 0.033}$ | $\mathbf{0.850 \pm 0.006}$ | $\mathbf{0.756 \pm 0.028}$ |

| WeakS-to-Strong-5 | Acc w/o aux_loss | PGR w/o aux_loss | Acc w/ aux_loss | PGR w/ aux_loss  |
|-|-|-|-|-|
| Naive Multi-Weak | 0.832 $\pm$ 0.005 | 0.641$\pm$ 0.025 | 0.853 $\pm$ 0.006 | 0.754 $\pm$ 0.032  |
| FlyingSquid | $\mathbf{0.832 \pm 0.004}$ | $\mathbf{0.643 \pm 0.023}$ | 0.855 $\pm$ 0.007 | 0.768 $\pm$ 0.035  |
| Bayesian Multi-Weak | 0.831 $\pm$ 0.008 | 0.627 $\pm$ 0.027 | $\mathbf{0.866 \pm 0.006}$ | $\mathbf{0.828 \pm 0.038}$   |

As shown in the tables, the FlyingSquid performs consistently worse than Naive Multi-Weak with three weak models, overall worse than Naive Multi-Weak when five weak models were used. Neither surpasses our proposed Bayesian method. The reason might be that five weak labels allow FlyingSquid to model label relationships more comprehensively, while simply averaging labels (naive approach) loses information. It doesn't outperform Bayesian because FlyingSquid is designed for hard labels and can't handle the confidence of weak models.

### G.2 Additional experiments on another dataset

We experimented on another dataset, CosmosQA[2], for the classification task. CosmosQA is a large-scale dataset of problems that require commonsense-based reading comprehension. Like the preprocess on the SciQ dataset, 5k data samples were extracted for training weak models, another 5k samples for strong models, and 1k samples for testing. This dataset is also reorganized into a binary classification format (i.e., determining correctness). Due to computational and time constraints, we only conducted experiments with three weak models. The results are listed below:

| Train on groundtruth | Accuracy  |
|-|-|
| GPT2-Large | 0.636  |
| OPT-1.3B | 0.642  |
| Pythia-1.4B | 0.654  |
| Qwen-7B | 0.847   |

*Table 1: Performance of single model on CosmosQA dataset. Trained on ground-truth labels.*

| Single Weak Model | Acc w/o aux_loss | PGR w/o aux_loss | Acc w/ aux_loss | PGR w/ aux_loss  |
|-|-|-|-|-|
| GPT2-Large | 0.652 $\pm$ 0.007 | 0.077 $\pm$ 0.032 | 0.651 $\pm$ 0.007 | 0.073 $\pm$ 0.032  |
| OPT-1.3B | 0.687 $\pm$ 0.005 | 0.218 $\pm$ 0.022 | 0.704 $\pm$ 0.009 | 0.304 $\pm$ 0.044  |
| Pythia-1.4B | 0.685 $\pm$ 0.010 | 0.162 $\pm$ 0.052 | 0.731 $\pm$ 0.002 | 0.399 $\pm$ 0.009   |

*Table 2: Weak-to-Strong performance with single weak model.*

| WeakS-to-Strong-3 | Acc w/o aux_loss | PGR w/o aux_loss | Acc w/ aux_loss | PGR w/ aux_loss  |
|-|-|-|-|-|
| Naive Multi-Weak | 0.691 $\pm$ 0.009 | 0.232 $\pm$ 0.043 | 0.698 $\pm$ 0.007 | 0.267 $\pm$ 0.034  |
| FlyingSquid | $\mathbf{0.699 \pm 0.004}$ | $\mathbf{0.272 \pm 0.020}$ | 0.706 $\pm$ 0.004 | 0.303 $\pm$ 0.030  |
| Bayesian Multi-Weak | 0.694 $\pm$ 0.007 | 0.247 $\pm$ 0.033 | $\mathbf{0.760 \pm 0.005}$ | $\mathbf{0.571 \pm 0.025}$   |

*Table 3: WeakS-to-Strong performance with three weak models.*

From the results, neither baseline outperformed the best single model (Pythia-1.4B with auxiliary loss), while our Bayesian approach significantly exceeded it, showing the effectiveness of our method, especially when weak model performance varies.

[1] Fu, D., Chen, M., Sala, F., Hooper, S., Fatahalian, K., & Ré, C.. Fast and three-rious: Speeding up weak supervision with triplet methods. In Proc. ICML. 2020.

[2] Huang, L., Le Bras, R., Bhagavatula, C., & Choi, Y.. Cosmos QA: Machine Reading Comprehension with Contextual Commonsense Reasoning. In Proc. EMNLP. 2019.

---

### Author Response · Authors · 2024-11-25
**Revised version of the manuscript**

We thank the reviewers for providing valuable suggestions and engaging in the discussion. We have updated the manuscript accordingly with the following revisions:
- Added an additional baseline and one more dataset (Section 5.2, 6.1, Appendix D).
- Motified explanation for the formulas for clarification (Section 3.3, 4.1).
- Added the specific value of weak models' weights $\lambda$ (Appendix F).

All changes are shown in blue. We appreciate reviewers' time and effort, and hope the revisions strengthen the paper.

---

### Author Response · Authors · 2024-12-03
**Summary of Responses**

Dear all reviewers, Area Chair, and Senior Area Chairs,

We thank reviewers WJwn, zdng, JkWZ, grX8 for their valuable feedback on our work. Below is a summary of responses to the reviewers.

### Reviewer WJwn

- **Baselines and Tasks**: We added a new baseline FlyingSquid, as well as a new dataset CosmosQA, for classification. The results are listed in a general response, demonstrating the effectiveness and generalization of our method.
- **Setting of Weight $\lambda$**: The specific value of $\lambda$ is informed by weak-to-strong performance, which is a direct way to reflect the quality of weak labels. No direct mapping relationship and grid search is done for selecting $\lambda$ weight.
- **Extension to Humans**: The scenarios and capabilities of weak models and humans are different, especially in evaluation capabilities. We designed some methods (for example, per-token probability estimation) to address this issue. When extending to humans, they can provide detailed, fine-grained evaluations across various aspects.

### Reviewer zdng

- **Datasets**: We chose the SciQ dataset for classification to follow Burn et al.'s work, and added experiments on another dataset CosmosQA. For the generation task, the SLURP dataset is chosen because of its clear evaluation metrics, making it easy to quantify the Weak-to-Strong performance.
- **Design choices for probability estimation**: Our motivation is that weak model performance is not satisfactory, and soft labels can provide more information than hard labels. Detailed explanation is given in response to the reviewer.

### Reviewer JkWZ

- **Motivation for Dirichlet**: The Dirichlet distribution is the conjugate prior distribution of the categorical distribution requested by both text classification and generation, which is tractable and avoids mathematical difficulties.
- **Training Stability**: Our strong model is not trained from scratch but is fine-tuned from a pre-trained model. Its inherent capability allows it to produce reliable confidence score estimates. We didn't encounter instability issues in our experiments.
- **Capability of Baselines**: For classification, we added FlyingSquid as another baseline. For generation, we introduced Joint Decoding, which is specifically designed for multiple weak model generating. The details can be found in response to the reviewer.

### Reviewer grX8

- **Model Size and Complementary**: We performed additional experiments with settings of the 1B weak + 13B strong models on SciQ dataset, as well as 3B weak + 13B strong models on both SciQ and CosmosQA datasets. The results show that our method is not tied to model size but to model performance. The results also give comprehensive insights into the complementary claim about how our method works.
- **Auxilary Loss**: Our Bayesian Multi-Weak method incorporates a distinct auxiliary loss design compared to baseline methods. The final approach combines both the Bayesian Multi-Weak framework and the auxiliary loss. The results without the auxiliary loss are presented solely for ablation studies.

---

### Meta-Review · Area_Chair_bgGD · 2024-12-23

**Metareview:**

This paper seeks to improve weak-to-strong generalization. This phenomenon, which has been known for quite a long time, has recently become popularized due to its theorized application to superalignment. In weak-to-strong generalization, a weak model produces outputs that a strong model is trained on. In certain settings, this strong model can perform better than the weak model did, despite only being trained on the weak model’s outputs.

The authors note that one challenge is the choice of the weak model. They propose replacing a single model with a simple Bayesian ensembling approach and show how to use this for both predictive and generative tasks. The latter setting is challenging, since it requires ensembling output sequences from multiple generators. They also show some nice extensions for alignment.

The reviewers generally liked the paper, with one caveat being the strength of the empirical results. While I share some of the sentiment, the paper has a bunch of good ideas and sufficient evidence of their value, so I voted for acceptance.

**Additional Comments On Reviewer Discussion:**

The major concern from the most negative reviewer was on scaling the approach to larger models. Although I can understand the request, I downweighted this concern for practical reasons; it is always possible to ask for additional evidence at larger scales, and the authors' choice of experiment seemed fine. Overall, I felt the authors provided a reasonable response to all of the questions brought up.

---

### Decision · Program_Chairs · 2025-01-22

Accept (Poster)